



# Validation of tropospheric NO₂ column measurements of GOME-2A and OMI using MAX-DOAS and direct sun network observations

Gaia Pinardi[1], Michel Van Roozendael[1], François Hendrick[1], Nicolas Theys[1], Nader Abuhassan[2,16], Alkiviadis Bais[3], Folkert Boersma[4,17], Alexander Cede[2,14], Jihyo Chong[5], Sebastian Donner[6], Theano Drosoglou[3], Udo Frieß[7], José Granville[1], Jay R. Herman[2,16], Henk Eskes[4], Robert Holla[8], Jari Hovila[9], Hitoshi Irie[10], Yugo Kanaya[3], Dimitris Karagkiozidis[3], Natalia Kouremeti[3,18], Jean-Christopher Lambert[1], Jianzhong Ma[11], Enno Peters[5], Ankie Piters[9], Oleg Postylyakov[12], Andreas Richter[5], Julia Remmers[6], Hisahiro Takashima[13,19], Martin Tiefengraber[14,20], Pieter Valks[15], Tim Vlemmix[4], Thomas Wagner[6], Folkard Wittrock[5]

[1]Royal Belgian Institute for Space Aeronomy, BIRA-IASB, Brussels, Belgium
[2]NASA/Goddard Space Flight Center, GSFC, Greenbelt, MD, USA
[3]AUTH, Aristotle University of Thessaloniki, Thessaloniki, Greece
[4]Royal Netherlands Meteorological Institute, KNMI, De Bilt, The Netherlands
[5]Gwangju Institute of Science and Technology GIST, Gwangju , Korea
[5]Institut für Umweltphysik, Universität Bremen, Bremen, Germany
[6]Max Planck Institute for Chemistry, Mainz, Germany
[7]Institut für Umweltphysik, Universität Heidelberg, Heidelberg, Germany
[8]German Weather Service, DWD, Hohenpeissenberg, Germany
[9]Finnish Meteorological Institute, FMI, Helsinki, Finland
[10]Center for Environmental Remote Sensing, Chiba University,Chiba, Japan
[11]Chinese Academy of Meteorological Sciences,Beijing, China
[12]A.M.Obukhov Institute of Atmospheric Physics, Russian Academy of Sciences, IAP/RAS, Moscow, Russia
[13]Research Institute for Global Change, JAMSTEC, Yokohama, Japan
[14]LuftBlick, Innsbruck, Austria
[15]Deutsches Zentrum für Luft- und Raumfahrt (DLR), Institut für Methodik der Fernerkundung (IMF), Oberpfaffenhofen, Germany
[16]University of Maryland, Joint Center for Earth Systems Technology, Baltimore, MD, USA
[17]Wageningen University, Wageningen, The Netherlands
[18]Physikalisch-Meteorologisches Observatorium Davos, World Radiation Center (PMOD/WRC), Davos Dorf, Switzerland
[19]Faculty of Science, Fukuoka University, Fukuoka Japan
[20]Department of Atmospheric and Cryospheric Sciences, University of Innsbruck, Innsbruck, Austria

*Correspondence to*: Gaia Pinardi (gaia.pinardi@aeronomie.be)

**Abstract**. MAX-DOAS and direct sun NO₂ vertical column network data are used to investigate the accuracy of tropospheric NO₂ column measurements of the GOME-2 instrument on the MetOP-A satellite platform and the OMI instrument on Aura. The study is based on 23 MAX-DOAS and 16 direct sun instruments at stations distributed worldwide. A method to quantify and correct for horizontal dilution effects in heterogeneous NO₂ field conditions is proposed. After systematic application of this correction to urban sites, satellite measurements are found to present smaller biases compared to ground-based reference data in almost all cases. We investigate the seasonal dependence of the validation results, as well as the impact of using different approaches to select satellite ground pixels in coincidence with ground-based data. In optimal comparison conditions (satellite pixels containing the station) the median bias between satellite tropospheric NO₂ column measurements and the ensemble of MAX-DOAS and direct sun measurements is found to be significant and equal to -36% for GOME-2A and -20% for OMI. These biases are further reduced to -24% and -8% respectively, after application of the dilution correction.



Comparisons with the QA4ECV satellite product for both GOME-2A and OMI is also performed, showing less scatter but also a slightly larger median tropospheric $NO_2$ column bias with respect to the ensemble of MAX-DOAS and direct sun measurements.

## 1 Introduction

Nitrogen dioxide ($NO_2$) is a key species for atmospheric chemistry, present both in the stratosphere and in the troposphere. In the troposphere, nitrogen oxides ($NOx = NO+NO_2$) together with volatile organic compounds are key ingredients for ozone and photochemical smog formation in polluted regions. By reaction with the hydroxyl radical (OH), $NO_2$ forms nitric acid ($HNO_3$) which leads to acid rain and consequently acidifies soils and water bodies with negative impacts on the environment. In addition to its important role in air quality (human health and environmental acidification), $NO_2$ is also relevant for climate processes at high concentrations, contributing to direct radiative forcing and the extension of atmospheric lifetimes of gases such as $CH_4$. The main sources of NOx include anthropogenic and natural emissions, such as fossil fuel combustion, biomass burning, lightning and microbial soil emissions. There is a need for accurate $NO_2$ measurements, to assess and forecast its impact on air quality.

$NO_2$ can be measured by several methods, such as in-situ sampling and active or passive remote sensing. The Differential Optical Absorption Spectroscopy (DOAS) technique (Platt and Stutz, 2008) is widely used to retrieve $NO_2$ in the atmosphere from measurements taken from satellites, balloons and from the ground. Since mid of the nineties, $NO_2$ is measured from space by mid-morning low earth orbit (LEO) nadir satellite instruments, such as GOME on ERS-2 (1996-2003, Burrows et al. (1999)), SCIAMACHY on ENVISAT (2002-2012, Bovensmann et al. (1999)) and GOME-2 on MetOp A, B and C (since 2006, 2012 and November 2018 respectively, Munro et al. (2016)). From 2004 onwards, $NO_2$ measurements in the early afternoon are also performed from the OMI imaging spectrometer on the EOS-Aura platform (Levelt et al. (2006)) and since the end of 2017 from the Sentinel-5p TROPOMI instrument (Veefkind et al. 2012). In the last 15 years, ground-based MAX-DOAS (Multi-Axis Differential Optical Absorption Spectroscopy) instruments have been developed to measure tropospheric trace gases (Hönninger and Platt, 2002; Hönninger et al., 2004; Sinreich et al., 2005). Combined with profiling algorithms, this technique has been successfully applied to retrieve tropospheric columns and information on the vertical distribution of $NO_2$, HCHO, $SO_2$, BrO, IO, HONO, CHOCHO and aerosols (e.g. Bobrowski et al., 2003; Wittrock et al., 2004; Wagner et al., 2004; Heckel et al., 2005; Frieß et al., 2006 and 2015; Sinreich et al., 2007; Theys et al., 2007; Irie et al., 2008, 2009; Clémer et al., 2010; Galle et al., 2010; Hendrick et al., 2014). Direct sun observations in the UV-visible, which provide total column measurements (Cede et al., 2006; Herman et al., 2009; Wang et al., 2010), are also used for monitoring atmospheric $NO_2$. In particular, the recently-developed Pandora instrument (SciGlob, http://www.sciglob.com/) provides operationally direct sun measurements of $O_3$, $NO_2$, $SO_2$ and HCHO (Herman et al., 2009; Wang et al., 2010; Tzortziou et al., 2015; Fioletov et al., 2016; Spinei et al., 2018; Herman et al., 2018; 2019) at a growing number of sites.

One of the strengths of LEO nadir satellite instruments with wide swath width, like OMI and GOME-2, is their daily global coverage. Their main drawback is their limited revisit frequency and associated sampling of the diurnal



cycle (typically one overpass per day for mid-latitudes) and coarse spatial resolution (from a few to several hundreds of kilometers). The accuracy of the different satellite datasets is also of concern, e.g. for trend analysis or diurnal variation studies. Validation activities, which are an essential part of any satellite programme, aim at deriving independently a set of indicators characterizing the quality of the data product. They encompass the

monitoring of instrumental stability as well as the inter-sensor consistency needed to ensure continuity between different satellite missions. Satellite validation also contributes to the improvement of retrieval algorithms through investigation of the accuracy of the data products and their sensitivity to retrieval parameter choices. Tropospheric satellite data products depend on various sources of ancillary data, e.g. a-priori vertical distribution of the absorbing and scattering species, surface albedo, information on clouds and aerosols (Boersma et al., 2004; Lin et al., 2015;

Lorente et al., 2017, Liu et al., 2019a). In the case of $NO_2$, separation between stratospheric and tropospheric contributions is an additional source of complexity in the retrieval, and there is considerable debate on the importance of the role of free tropospheric (background) $NO_2$ in the retrieval process (Jiang et al., 2018; Silvern et al., 2019). As discussed by Richter et al. (2013), the validation of tropospheric reactive gases (such as $NO_2$, HCHO and $SO_2$) is also challenging because short atmospheric lifetimes, local emission sources and transport can

lead to a large variability of their concentrations in time and space (both vertically and horizontally). Active photochemistry and transport processes lead to important diurnal variations cycles (Boersma et al., 2008) that need to be considered for validation studies. MAX-DOAS and direct sun remote-sensing techniques have large potential capacities for the validation of satellite trace gas observations, as they measure all day long and provide accurate measurements of integrated column amounts (i.e. a quantity close to that measured by space-borne instruments).

MAX-DOAS and direct sun measurements also match better the horizontal resolution of satellite observations than e.g. surface in-situ monitoring networks. The spatial averaging of MAX-DOAS measurements has been quantified and shown to range from a few km to tens of km depending on aerosol content and measurement wavelength (Irie et al., 2011, 2012; Wagner et al., 2011; Wang et al., 2014; Gomez et al., 2014; Ortega et al., 2015).

In the last decade, several studies compared different SCIAMACHY, GOME-2 and OMI $NO_2$ data products (generated by both operational and scientific prototype processors) to MAX-DOAS measurements at various stations (e.g., Brinksma et al., 2008; Hains et al., 2010; Vlemmix et al., 2010; Irie et al., 2008; Ma et al., 2013; Lin et al., 2014; Wang et al. 2017b; Drosoglou et al., 2017, 2018; Liu et al., 2019a, b, c). JAMSTEC data from the

MADRAS network have been used in Kanaya et al. (2014) for the validation of the OMI DOMINO and NASA tropospheric $NO_2$ data. BIRA-IASB MAX-DOAS stations have been regularly used for the validation of GOME-2 GDP products from MetOp-A and MetOp-B (Valks et al., 2011; Pinardi et al., 2011; 2013; 2015; Liu et al., 2019b) as part of the AC SAF activities (Hassinnen et al., 2016; see also [www.cdop.aeronomie.be/validation/valid-results](http://www.cdop.aeronomie.be/validation/valid-results)). Pandora datasets have also been used in satellite validation of total and tropospheric $NO_2$ columns

(Herman et al. 2009; Tzortziou et al., 2014; 2015; Judd et al., 2019) and a recent study of Herman et al. (2019) presented an overview at 14 Pandora sites showing that NASA OMI $NO_2$ overpass data consistently underestimate the Pandora derived $NO_2$ amounts. One general conclusion of these exercises was to find a low bias of the satellites tropospheric $NO_2$ columns in urban conditions and, in contrast, a better agreement with ground-based data in background and pristine locations (Celarier et al., 2008; Halla et al., 2011; Kanaya et al., 2014). However Irie et





al. (2012) also reported low OMI $NO_2$ column values over China in summer, when the spatial distribution of $NO_2$ was likely homogeneous.

In the present study, we validate GOME-2A and OMI tropospheric $NO_2$ column measurements using data from a large number of MAX-DOAS and direct sun instruments operating in Europe, Asia, North America and Africa under a wide variety of atmospheric conditions and pollution patterns. Some of these datasets have already been used in the past for tropospheric $NO_2$ validation of different satellites and products. In the present study we combine them in a coordinated way allowing for a global approach to satellite validation, sampling different $NO_2$ levels in various locations around the globe. In addition the smearing (or dilution) of the $NO_2$ field due to the limited horizontal resolution of satellite measurements is investigated. A method for the quantification and correction of the dilution effect is proposed, and its impact on validation results is quantitatively evaluated. Our validation approach is applied to operational OMI DOMINO and AC SAF GOME-2A products, as well as to climate data record OMI and GOME-2A $NO_2$ data products generated within the EU QA4ECV project.

The paper is structured as follows: Sections 2 and 3 describe the OMI and GOME-2A sensors and data sets as well as the reference ground-based measurements. Section 4 presents the comparison methodology and comparison results are discussed in Section 5. In Section 6, we concentrate on the quantification of horizontal dilution effects in satellite measurements performed around the measurement sites, and we show how these effects impact the validation results in urban conditions. Section 7 addresses seasonal effects and the impact of satellite data selection on the comparison results. Section 8 presents a summary of the validation results, and conclusions are detailed in Section 9.

## 2. Satellite tropospheric $NO_2$ datasets

Tropospheric $NO_2$ data products from space-borne sensors are generally retrieved via three main steps. First, a DOAS spectral analysis yielding the total column amount of $NO_2$ along the slant optical path, secondly an estimation of the stratospheric $NO_2$ column, to be subtracted from the total column to derive the tropospheric contribution (so-called residual technique), and finally a conversion from slant (SCD) to vertical columns (VCD). The last step is based on air-mass factor (AMF) calculations which require a-priori knowledge of the $NO_2$ vertical distribution, pressure and temperature, surface albedo, aerosols and information on (effective) cloud cover and height (Boersma et al., 2004). The retrieval of tropospheric $NO_2$ is given by:

$$VCDtropo = \frac{(SCD-AMFstrato*VCD\ strato)}{AMFtropo} \tag{1}$$

Different data products have been generated for each satellite instrument, using different assumptions for each of the three aforementioned steps (see Boersma et al. 2004; Richter et al, 2011; Lin et al., 2014; Bucsela et al., 2013; Lamsal et al., 2014; van Geffen et al., 2015; Krotkov et al., 2016; Lorente et al., 2017; Liu et al., 2019a,b,c). In addition to instrument-specific differences, structural uncertainties arising from the application of different retrieval methodologies to the same satellite observations (sometimes also called forward model uncertainties), can introduce differences in the retrieved tropospheric $NO_2$ columns ($VCD_{tropo}$) of 10-50% (e.g. van Noije et al., 2006; Lorente et al., 2017; Zara et al., 2018). Slant column densities (SCD) structural uncertainties generally do not exceed $1 \times 10^{15}$ molec/$cm^2$, while the AMF calculation leads to more significant uncertainties (Boersma et al.,



2004) which can be separated into implementation differences (when different groups use identical ancillary data for the calculation of tropospheric $NO_2$ AMFs) of about 6%, and structural differences, due to ancillary data selection, which can reach 31-42% (Lorente et al, 2017). The uncertainty in separating the stratospheric and tropospheric columns is about $0.5 \times 10^{15}$ molecules/cm$^2$ (Dirksen et al., 2011; Lorente et al., 2017).

In the present study, we focus on the ground-based validation of the mid-morning GOME-2A and the early afternoon OMI data. Illustration of the validation method and step-by-step results along the manuscript are given for the GOME-2A GDP (GOME Data Processor) 4.8 $NO_2$ operational data product (Valks et al. 2011) and the OMI DOMINO v2.0 data product (Boersma et al., 2011), while final validation results and discussion also gather

results for the GOME-2A and OMI QA4ECV products (Boersma et al., 2018; Zara et al., 2018). All products are briefly presented in Table 1 and in the following sub-sections.

**2.1 GOME-2 products**

The second Global Ozone Monitoring Instrument (GOME-2) is a nadir-looking UV-visible spectrometer

measuring the solar radiation backscattered by the atmosphere and reflected by the Earth and clouds in the 240–790 nm wavelength interval and with a spectral resolution of 0.2–0.5 nm full width at half maximum (FWHM) (Munro et al., 2016). There are three versions of GOME-2 instruments flying on a Sun-synchronous polar orbit on board the Meteorological Operational satellites (MetOp-A, MetOp-B and MetOp-C, launched respectively in October 2006, September 2012, and November 2018). They have an Equator crossing time of 09:00-09:30 local

time in the descending node. In this study we concentrate on the GOME-2A instrument (that is, on MetOp-A), which presents the longest data record. The default swath width of the GOME-2A across-track scan is 1920 km, allowing global Earth coverage within 1.5–3 days at the Equator, with a nominal ground pixel size of 80×40km$^2$. Since 15 July 2013, GOME-2A is measuring on a reduced swath mode of 960km, with a ground pixel size of 40×40km$^2$.

Operational products are retrieved from GOME-2 measurements in the framework of the Atmospheric Composition Satellite Application Facility AC SAF (www.acsaf.fmi.fi, formerly O3M SAF; see also Hassinnen et al., 2016). Total, tropospheric and stratospheric $NO_2$ columns are operationally retrieved with the GOME Data Processor (GDP) and a description of this algorithm can be found in Valks et al. (2011) and Liu et al. (2019b).

Within the QA4ECV (Quality Assurance for Essential Climate Variables) project, a coherent offline $NO_2$ dataset has been created for GOME, SCIAMACHY, GOME-2A and OMI (Boersma et al., 2018; Zara et al., 2018; Lorente et al, 2017) and comparisons with this dataset are also included at the end of this study.

Table 1 summarizes the main retrieval steps for the various tropospheric $NO_2$ products considered here. Main

differences are related to the methods to obtain the stratospheric $NO_2$ column, the cloud parameters and the a-priori information used to calculate the tropospheric air mass factor. In the Q4ECV case, stratospheric columns are derived using two different approaches (assimilation in TM4 and STREAM). The stratospheric separation method has an estimated uncertainty in the $0.15–0.3 \times 10^{15}$ molec/cm$^2$ range (Valks et al., 2011). The typical overall





uncertainty for individual retrievals of tropospheric $NO_2$ vertical column densities is estimated to be $1.0 \times 10^{15}$ molecules/cm$^2$ ($\pm 25\%$) in rural environments and from 40% to 80% under polluted conditions (Valks et al., 2011).

Previous validation of GOME-2A GDP 4.8 data can be found in Valks et al. (2011); Hassinnen et al. (2016); Liu

et al. (2018b) for a few MAX-DOAS stations, and results of regular validation exercises can be found on www.cdop.aeronomie.be/valid-results. Satellite-to-satellite comparisons of the GOME-2A QA4ECV data has been performed by Zara et al. (2018), Lorente et al. (2018) and Liu et al. (2019b). Previous GOME-2 validation highlighted the effect of GOME-2 large pixels, and the aerosol shielding effect, leading e.g., to differences of 5% to 25% over China (Ma et al., 2013; Wu et al., 2013; Wang et al., 2017; Drosoglou et al., 2018). Liu et al. (2019b)

showed possible improvements of the GDP 4.8 product, leading to reduced discrepancies of the satellite-to-ground-based biases of the order of 10% to 25% for several MAX-DOAS stations.

### 2.2 OMI products

OMI (Ozone Monitoring Instrument) is a nadir-viewing imaging spectrometer with a spectral resolution of about

0.5 nm FWHM (Levelt et al., 2006). The light entering the telescope is depolarized using a scrambler and split into two spectral bands: a UV channel (wavelength range 270–380 nm) and a visible channel (wavelength range 350–500 nm). The 114° viewing angle of the telescope corresponds to a 2600km wide swath on the Earth's surface distributed over 60 cross-track positions, which enables quasi-global coverage in one day. In the nominal global operation mode, the OMI ground pixel size varies from 13×24km² at true-nadir to 28×150km² on the edges of the

swath. OMI is onboard the EOS-Aura satellite that was launched in July 2004, in a sun-synchronous polar orbit crossing the Equator around 13:30 LT (in ascending node). The radiometric stability of the OMI instrument is exceptionally good (Schenkeveld et al., 2017), however, since June 2007, several rows of the detector have been affected by a signal reduction, the so called "row anomaly" (http://www.knmi.nl/omi/research/product/rowanomaly-background.php), reducing the usable swath coverage

(see Boersma et al., 2018).

The DOMINO (Derivation of OMI tropospheric $NO_2$) product is distributed in NRT via the TEMIS (Tropospheric Emission Monitoring Internet Service, http://www.temis.nl) project (Boersma et al., 2011). The offline OMI QA4ECV v1.1 product (Boersma et al., 2018), is very similar to the GOME-2 product, as can be seen in Table 1.

For OMI, the stratospheric separation is performed using a data assimilation scheme based on the TM4 or TM5-MP chemistry transport models. Its uncertainty is estimated to be about $0.2–0.3 \times 10^{15}$ molec/cm$^2$ (Boersma et al., 2004; Dirksen et al., 2011). Stratospheric $NO_2$ vertical columns used in our study are derived from assimilated stratospheric slant columns divided by a geometrical air-mass factor, as described in Hendrick et al. (2012). For the OMI QA4ECV dataset, two estimates of the stratospheric column are reported (data assimilation and

STREAM), and Boersma et al. (2018) has illustrated the differences for both approaches, with differences up to $1 \times 10^{15}$ molec/cm$^2$. Compernolle et al. (2020) showed best agreement with ZSL-DOAS NDACC measurements for the STREAM stratospheric dataset, with mean differences between the 2 datasets of the order of $0.2 \times 10^{15}$ molec/cm$^2$ on average.



OMI DOMINO v2.0 has been widely used in the past, and several validation exercises (Brinksma et al., 2008; Hains et al., 2010; Vlemmix et al., 2010; Irie et al., 2008, 2012; Lin et al., 2014; Wang et al. 2017b; Drosoglou et al., 2017, 2018; Liu et al. 2019a) found underestimation of the OMI tropospheric $NO_2$ columns in urban conditions and a better agreement in background locations (Celarier et al., 2008; Halla et al., 2011; Kanaya et al., 2014).

Kanaya et al. (2014) showed close correlations with MAX-DOAS observations at 7 stations, but found low biases up to ~50 %. Regarding the OMI QA4ECV product, Boersma et al. (2018) reported a first validation at the Tai'an station (China) in one summer month finding good agreement (bias of -2 %) with respect to MAX-DOAS $NO_2$ columns (better than the agreement found for DOMINO v2 of -11% bias). Liu et al. (2019a) investigated the impact of correcting for aerosol vertical profiles in the OMI data, and compared four OMI datasets (POMINO and

POMINO v1.1, DOMINO v2.0 and QA4ECV) with respect to data of three Chinese stations. Results suggested a significant improvement of the OMI $NO_2$ retrieval when correcting for aerosol profiles, in general and for hazy days. This is consistent with the previous finding that the accuracy of DOMINO v2.0 is reduced for polluted, aerosol-loaded scenes (Boersma et al., 2011; Kanaya et al., 2014; Lin et al., 2014; Chimot et al., 2016). Liu et al, also showed discrepancies in DOMINO v2.0 for very high $NO_2$ values (> 70 x$10^{15}$ molec/cm$^2$), For 18 cloud-free

days, they found smaller differences between the four products with respect to MAX-DOAS, with the QA4ECV dataset having the highest $R^2$ (0.63) and the lowest bias (-5,8 %). An extended validation of the QA4ECV OMI product is reported in the recent Compernolle et al. (2020) study, showing a negative bias (from −1 to −4 x$10^{15}$ molec/cm$^2$) with respect to 10 MAX-DOAS instruments, a feature also found for the OMI OMNO2 standard data product. They also found that the tropospheric VCD discrepancies between satellite and ground-based data exceed

the combined measurement uncertainties and that, depending on the site, this discrepancy could be attributed to a combination of comparison errors (horizontal smoothing difference error, error related to clouds and aerosols and differences due to a priori profile assumptions).



**Table 1**: Description of the satellite retrievals algorithms involved in this study.

| | GOME-2A | | OMI | |
|---|---|---|---|---|
| **Instrument information** | | | | |
| **Resolution at nadir (across x along track)** | 80x40 km² * | | 24x13 km² | |
| **Solar Local Time at Equator crossing node** | 9h30 | | 13h30 | |
| **NO₂ retrieval information** | | | | |
| **Version** | GDP 4.8 | QA4ECV v1.1 | QA4ECV v1.1 | DOMINO v2.0 |
| **Reference** | Valks et al., 2011; 2017 | Boersma et al., 2018, Zara et al.2018 | Boersma et al., 2018, Zara et al.2018 | Boersma et al., 2011 |
| **SCD retrieval** | DOAS fitting window: 425-450nm Absorbers: NO₂, O₃, O₂-O₂, H₂O and Ring | DOAS fitting window: 405-465nm Absorbers: NO₂, O₃, O₂-O₂, H₂O, H₂Oliq and Ring | DOAS fitting window: 405-465nm Absorbers: NO₂, O₃, O₂-O₂, H₂O, H₂Oliq and Ring | DOAS fitting window: 405-465nm Absorbers: NO₂, O₃, H₂O and Ring |
| **Stratospheric Correction** | Spatial filtering/masking of polluted fields | - Assimilated NO₂ stratospheric slant columns with the TM5-MP (selected as default) - STREAM (Beirle et al., 2016) | - Assimilated NO₂ stratospheric slant columns with the TM5-MP (selected as default) - STREAM (Beirle et al., 2016) | Assimilated NO₂ stratospheric slant columns with the TM4 chemistry-transport model |
| **Tropospheric AMF calculation** | | | | |
| **- Radiative Transfer Model** | LIDORT | DAK 3.0 | DAK 3.0 | DAK 3.0 |
| **- NO₂ a-priori profile** | Monthly profiles for 1997 from MOZARTv2 (Horowitz et al., 2003), 1.875°x1.875° resolution | Daily profiles from TM5-MP model (Williams et al., 2017), 1°x1° resolution | Daily profiles from TM5-MP model (Williams et al., 2017), 1°x1° resolution | Daily profiles from TM4 model (Huijnen et al., 2010), 2°x3° resolution |
| **- Cloud treatment** | IPA correction based on OCCRA/ROCINN cloud scheme v3 (Loyola et al., 2017) | IPA correction based on FRESCO+ cloud algorithm (Wang et al., 2008) | IPA correction based on OMCLDO2 cloud algorithm (Veefkind et al., 2016) | IPA correction based on OMCLDO2 cloud algorithm (Acarreta et al., 2004; Stammes et al., 2008) |
| **- aerosol** | Implicitly corrected by cloud treatment | Implicitly corrected by cloud treatment | Implicitly corrected by cloud treatment | Implicitly corrected by cloud treatment |
| **- Albedo** | 1.25° lon×1° lat surface LER climatology derived from combined | climatology from Tilstra et al. (2017), | updated 5-year climatology (Kleipool et al., 2008). | 0.5°x0.5° OMI climatology (Kleipool et al., 2008) |





| | TOMS/GOME measurements (Boersma et al., 2004) | | | |
|---|---|---|---|---|
| **Overall estimated uncertainty of tropospheric NO₂ vertical column densities** | $1.0\times10^{15}$ molecules/cm$^2$ ($\pm25\%$) in rural environments and from 40% to 80% under polluted conditions (Valks et al., 2011) | Average of 35% to 45% single pixel uncertainties in polluted regions (Boersma et al., 2018) | Average of 35% to 45% single pixel uncertainties in polluted regions (Boersma et al., 2018) | $1.0\times10^{15}$ molecules/cm$^2$ ($\pm25\%$) (Boersma et al., 2011; Lin et al., 2014; Lamsal et al., 2014). |

\* since 15 July 2013 GOME-2A operates in a reduced swath mode, corresponding to a ground pixel size of 40x40km²





**Table 2**: MAX-DOAS tropospheric NO$_2$ datasets included in this study (23 stations, 15 with profiles). GA stands for geometrical approximation, OEM for Optimal Estimation Method and PP for Parametrized Profiling.

| Station/Country (lat/long) | Station Type | Owner/ Group | Time Period | Instrument Type | Retrieval Type | Reference |
|---|---|---|---|---|---|---|
| Bremen/Germany, (53°N, 9°E) | Urban | IUPB | 01/2007- 08/2018 | Custom-built MAX-DOAS | VCD from QA4ECV | QA4ECV dataset |
| De Bilt /The Netherlands, (52.10°N, 5.18°E) | Urban | KNMI | 11/2007- 08/2018 | miniDOAS | VCD with fixed profile shape | Vlemmix et al., 2010 QA4ECV and NIDFORVAL datasets |
| Uccle/Belgium, (50.78° N, 4.35° E) | Urban | BIRA-IASB | 04/2011-02/2016 | miniDOAS | VCD and profiles from OEM | Gielen et al., 2014 |
| Mainz/Germany, (50°N, 8°E) | Urban | MPIC | 06/2013-08/2018 | Custom-built MAX-DOAS | VCD from QA4ECV | QA4ECV dataset |
| Thessaloniki/ Greece, (40.63°N, 22.96°E) | Urban | AUTH | 01/2011 – 08/2018 | Phaethon | VCD from QA4ECV | Kouremeti et al., 2013; Drosoglou et al., 2017 QA4ECV datasets |
| Beijing/China, (39.98°N, 116.38°E) | Urban | BIRA-IASB/IAP | 07/2008-04/2009 | Custom-built MAX-DOAS | VCD and profiles from OEM | Clémer et al., 2010; Hendrick et al., 2014; Vlemmix et al., 2015 |
| Beijing/China, (39.95°N, 116.32°) | Urban | CAMS | 08/2008-09/2011 | miniDOAS | VCD from GA at 30°elev | Ma et al., 2013 |
| Athens/Greece, (38.05°N, 23.86°E) | Urban | IUPB/NOA | 09/2012 - 08/2018 | Custom-built MAX-DOAS | VCD from QA4ECV | QA4ECV datasets |
| Chiba/Japan, (35.63°N, 140.10°E) | Urban | ChibaU | 06/2012 – 07/2017 | CHIBA-U MAX-DOAS | VCD and profiles from PP | Irie et al., 2011; Irie et al., 2012; Irie et al., 2015 ; Irie et al., 2019 |
| Yokosuka/Japan, (35.32°N, 139.65°E) | Urban | JAMSTEC | 10/2007-12/2015 | MADRAS MAX-DOAS | VCD and profiles from PP | Kanaya et al., 2014 |
| Gwangju/South Korea, (35.23°N, 126.84°E) | Urban | JAMSTEC | 01/2008-12/2015 | MADRAS MAX-DOAS | VCD and profiles from PP | Kanaya et al., 2014 |
| Nairobi/Kenya, (1°S, 36.50°E) | Urban | IUPB | 02/2011-12/2013 | Custom-built MAX-DOAS | VCD from QA4ECV | QA4ECV datasets |





| | | | | | | |
|---|---|---|---|---|---|---|
| Bujumbura/Burundi, (3°S, 29°E) | Urban | BIRA-IASB | 11/2013-07/2017 | Custom-built MAX-DOAS | VCD and profiles from OEM | De Smedt et al., 2015, Gielen et al., 2017 |
| Zvenigorod/Russia, (55.70°N, 36.78°E) | Sub-urban | JAMSTEC | 10/2008-12/2012 | MADRAS MAX-DOAS | VCD and profiles from PP | Kanaya et al., 2014 |
| Xianghe/China, (39.75° N, 116.96° E) | Sub-urban | BIRA-IASB | 03/2010 -08/2018 | Custom-built MAX-DOAS | VCD and profiles from OEM | Hendrick et al., 2014; Vlemmix et al., 2015 |
| Tsukuba/Japan, (36.05°N, 140.12°E) | Sub-urban | ChibaU | 01/2007 -04/2014 | CHIBA-U MAX-DOAS | VCD and profiles from PP | Irie et al., 2011; Irie et al., 2012 ; Irie et al., 2015 ; Irie et al., 2019 |
| Kasuga/Japan, (33.52°N, 130.48°E) | Sub-urban | ChibaU | 12/2013 – 07/2017 | CHIBA-U MAX-DOAS | VCD and profiles from PP | Irie et al., 2011; Irie et al., 2012 ; Irie et al., 2015 ; Irie et al., 2019 |
| Cabauw/The Netherlands (51.97°N, 4.93°E) | Remote | KNMI | 03/2011-08/2018 | miniDOAS | VCD from QA4ECV | QA4ECV and NIDFORVAL datasets |
| Hohenpeissenberg/ Germany, (47.80°N, 11.67°E) | Remote | IUPH/DWD | 05/2012-12/2012 | Custom-built MAX-DOAS | VCD and profiles from OEM | Yilmaz 2012, Niebling, 2010 |
| OHP/France, (43.94°N, 5.71°E) | Remote | BIRA-IASB | 02/2005 -12/2016 | Custom-built MAX-DOAS | VCD from QA4ECV | Valks et al., 2011 QA4ECV datasets |
| Fukue/Japan, (32.75°N, 128.68°E) | Remote | JAMSTEC | 03/2009-12/2015 | MADRAS MAX-DOAS | VCD and profiles from PP | Kanaya et al., 2014 |
| Cape Hedo/Japan, (26.87°N, 128.25°E) | Remote | JAMSTEC | 04/2007-12/2015 | MADRAS MAX-DOAS | VCD and profiles from PP | Kanaya et al., 2014 |
| Reunion LePort/ Reunion Island (20.9°S, 55.36°E) | Remote | BIRA-IASB | 4/2016-01/2018 | Custom-built MAX-DOAS | VCD and profiles from OEM | Theys et al., 2007 |

**Table 3**: Direct sun instruments measuring total $NO_2$ VCD included in this study (16 stations).

| Station/Country (lat/long) | Station Type | Owner/ Group | Time Period | Instrument Type | Reference |
|---|---|---|---|---|---|
| FMI, Helsinki/Finland, (60.20°N, 24.96°E) | Urban | NASA and FMI | 09/2011-06/2013 | Pandora | Herman et al., 2009; Tzortziou et al., 2014 |





| | | | | | |
|---|---|---|---|---|---|
| Harvard/USA, (42.67°N, 71.12°W) | Urban | NASA | 11/2014-08/2015 | Pandora | Herman et al., 2009; Tzortziou et al., 2014 |
| Thessaloniki/Greece, (40.63°N, 22.96°E) | Urban | AUTH | 01/2011 – 05/2014 | PHAETON (direct sun mode) | Kouremeti et al., 2013; Drosoglou et al., 2017 |
| Boulder/USA, (39.99°N, 105.26°W) | Urban | NASA | 12/2013-08/2015 | Pandora | Herman et al., 2009; Tzortziou et al., 2014 |
| Beijing/China, (39.98°N, 116.38°E) | Urban | BIRA-IASB | 07/2008-04/2009 | MAX-DOAS (direct sun mode) | Clémer et al., 2010; Hendrick et al., 2014; Vlemmix et al., 2015 |
| GSFC/USA, (38.99°N, 76.84°W) | Urban | NASA | 05/2009-08/2015 | Pandora | Herman et al., 2009; Tzortziou et al., 2014 |
| NASA HQ/USA, (38.88°N, 77.01°W) | Urban | NASA | 08/2012-08/2015 | Pandora | Herman et al., 2009; Tzortziou et al., 2014 |
| Seoul/South Korea, (37.59°N, 126.93°E) | Urban | NASA | 03/2012-8/2015 | Pandora | Herman et al., 2009; Tzortziou et al., 2014 |
| Busan/Korea, (35.24°N, 129.08°E) | Urban | NASA | 03/2012-05/2015 | Pandora | Herman et al., 2009; Tzortziou et al., 2014 |
| UHMT/USA, (29,72°N, 95.34°W) | Urban | NASA | 03/2012-04/2015 | Pandora | Herman et al., 2009; Tzortziou et al., 2014 |
| Xianghe/China, (39.75° N, 116.96° E) | Sub-urban | BIRA-IASB | 03/2010-08/2018 | MAX-DOAS (direct sun mode) | Hendrick et al., 2014; Vlemmix et al., 2015 |
| Langley/USA, (37.10°N, 76.39°W) | Sub-urban | NASA | 01/2010-06/2014 | Pandora | Herman et al., 2009; Tzortziou et al., 2014 |
| SERC/USA, (38.88°N, 76.55°W) | Remote | NASA | 09/2010-01/2013 | Pandora | Herman et al., 2009; Tzortziou et al., 2014 |
| Four Courner NM/USA, (36.80°N, 108.48°W) | Remote | NASA | 06/2012-07/2015 | Pandora | Herman et al., 2009; Tzortziou et al., 2014 |
| Izana/Spain, (28.31°N, 16.50°W) | Remote | NASA | 01/2013-08/2015 | Pandora | Herman et al., 2009; Tzortziou et al., 2014 |
| Mauna Loa/USA, (19.48°N, 155.60°W) | Remote | NASA | 11/2014-05/2015 | Pandora | Herman et al., 2009; Tzortziou et al., 2014; |



## 3. Ground-based datasets: MAX-DOAS and direct sun measurements

### 3.1 MAX-DOAS technique

A MAX-DOAS instrument measures the scattered sunlight under a sequence of viewing elevation angles extending from the horizon to the zenith (Fig. 1a). At low elevation angles, the observed sunlight travels a long path in the lower troposphere (under aerosol-free conditions, the lower the elevation angle, the longer the path) while all observations have approximately the same light path in the stratosphere, independently of viewing elevation. By taking the difference in SCD between off-axis observations and a (nearly) simultaneously acquired zenith reference spectrum (the differential slant column), the stratospheric contribution can therefore be eliminated. Tropospheric absorbers can be measured along the day, generally up to a solar zenith angle (SZA) of approximately 85° (Hönninger et al., 2004; Sinreich et al., 2005).

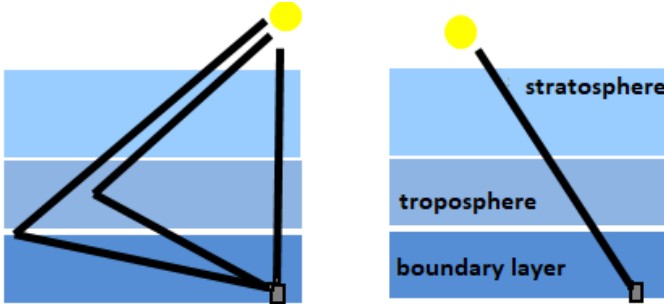

**Figure 1**: Sketches illustrating the MAX-DOAS and direct sun viewing geometries.

Radiance spectra acquired at different elevation angles are analyzed using the DOAS method (Platt and Stutz, 2008), which gives integrated trace gas concentrations along the atmospheric absorption path. The resulting differential slant columns (dSCDs) can be converted to vertical columns and/or vertical profiles using methods of different levels of complexity. Table 2 presents details about the retrieval strategy adopted by different teams. They generally belong to one of the following categories:

- Geometrical Approximation (GA): the vertical column is determined under the assumption that a single-scattering approximation can be made for moderately high elevation angles $\alpha$ (typically 30°) so that a simple geometrical air-mass factor ($AMF\alpha \equiv SCD/VCD = 1/\sin(\alpha)$) (Honninger et al., 2004; Brinksma et al., 2008; Ma et al., 2013) can be used,

- QA4ECV datasets: the vertical column is calculated using tropospheric AMFs based on climatological profiles and aerosol situations as developed during the QA4ECV project (http://uv-vis.aeronomie.be/groundbased/QA4ECV_MAXDOAS/QA4ECV_MAXDOAS_readme_website.pdf). These data are less sensitive to relative azimuth angle than the purely geometric approximation presented above.

- Vertical profile algorithms using an Optimal Estimation Method (OEM, Rodgers, 2000): these make use of a-priori vertical profiles and associated uncertainties (Friess et al., 2006; Clémer et al 2010; Hendrick et al., 2014; Wang et al., 2017b; Friedrich et al., 2019; Bösch et al., 2018),



- Vertical profile algorithms based on parameterized profile shape functions: these make use of analytical expressions to represent the trace gas profile using a limited number of parameters (Irie et al., 2008; 2011; Li et al., 2010; Vlemmix et al., 2010; Wagner et al., 2011; Beirle et al., 2019).

MAX-DOAS profile inversion algorithms use a two-step approach: in the first step, aerosol extinction profiles are retrieved from the measured absorption of the oxygen dimer $O_4$ (Wagner et al., 2004; Friess et al. 2006). In a second step, trace gas profiles are retrieved from the measured trace gas absorptions, taking into account the aerosol extinction profiles retrieved in the first step. Both OEM and parameterized profiling approaches provide vertical profiles of aerosols and $NO_2$ with a sensitivity typically in the 0-4 km altitude range with generally between 1.5

and 3 independent pieces of information in the vertical dimension (Vlemmix et al., 2015, Friess et al., 2016, Friess et al., 2019). This complementary information on the vertical distribution of gases and aerosols in the atmosphere has been used in some studies to test some key assumptions made in the satellite data retrieval, in particular the a-priori $NO_2$ profile and aerosols content, providing therefore more insight into the quality of the satellite data (e.g., Wang et al., 2017b; Liu et al., 2019b,c; Compernolle et al., 2020).Recent intercomparison studies (Vlemmix et al.,

2015; Friess et al., 2019; Tirpitz et al., 2020) show that both OEM and parameterized inversion approaches lead to consistent results in terms of tropospheric vertical column but larger differences in terms of profiles. In this study, every data provider submitted data retrieved with their own tools and formats, without any harmonization. Our study focuses therefore only on the vertical column, which is the more robust and reliable retrieved quantity. The time coverage of the different data sets used in this study is presented in Fig. S1.

The accuracy of the MAX-DOAS technique depends on the SCD retrieval noise, the uncertainty of the $NO_2$ absorption cross-sections and most importantly the uncertainty of the tropospheric AMF calculation. The estimated total error on $NO_2$ VCD is of the order of 7-17% in polluted conditions. This includes both random (around 3 to 10% depending on the instruments) and systematic (11 to 14%) contributions (e.g. Irie et al., 2008, 2011, 2012;

Wagner et al., 2011; Hendrick et al., 2014; Kanaya et al., 2014). In extreme cases, the error can however reach ~30% depending on geometry and aerosols.

**3.2 Direct sun technique**

Equipped with a sun-tracking device, direct sun instruments measure spectra of direct sun light having traversed

the whole atmosphere. Such instruments are sensitive to both troposphere and stratosphere (Figure 1b) and they provide accurate total column measurements with a minimum of a-priori assumptions.

Direct sun observations are routinely available from Pandora instruments. A standardized Pandora network has been set-up by NASA (Herman et al., 2009, Tzortziou et al., 2014) and extended by ESA and LuftBlick via the

Pandonia project (www.pandonia.net) to form the PGN (Pandonia Global Network, https://www.pandonia-global-network.org/). Pandora data used in this study originate mostly from the original NASA network, which includes more than 60 different sites covering different time-periods (mostly campaign-based). In total, 15 Pandora direct sun instruments delivering at least 3 months of data have been considered here. They are listed in Table 3 with an indication of their location, ownership, availability (see also Fig. S2) and references. Pandora instruments are





generally operated in polluted areas (urban or sub-urban), however the network also contains a few background/remote sites located in Europe, Asia and the US. Valid data were selected for normalized root-mean square of weighted spectral fitting residuals (wRMS) less than 0.005, uncertainty in $NO_2$ retrievals less than 0.05 DU were kept (A. Cede, personal communication).

Recent detailed studies in US and Korean sites during DISCOVER-AQ have shown good agreement of Pandora instruments with aircraft in-situ measurements, within 20% on average, although larger differences are observed for individual sites (Choi et al., 2019), the largest discrepancies being found in Texas (Nowlan et al., 2018). Good agreement of a few percent between Pandora and GeoTASO has been reported by Judd et al. (2019), while

differences increase when resampling the comparisons for larger simulated pixel sizes, up to about 40% bias for 18x18km², similar to the bias found with OMI (50%).

The Pandora spectrometers provide $NO_2$ total vertical column observations with a random uncertainty of about $2.7 \times 10^{14}$ molec/cm² and a systematic uncertainty of $2.7 \times 10^{15}$ molec/cm² (Herman et al., 2009). Those accounts for

DOAS fit systematic errors, random noise, and uncertainties related to the estimation of the residual gas amount in the reference spectra. In the present study, direct sun tropospheric VCDs are derived from the measured total $NO_2$ content after subtraction of the stratospheric part estimated using satellite data (alone or within assimilation scheme, see Sect. 2), interpolated to the geolocation of the Pandora spectrometer:

$$VCDtropo(DS) = VCDtot(DS) – VCDstrato(SAT) \qquad (2)$$

Summing the Pandora error uncertainty and the error uncertainty on the stratospheric column in quadrature, this approach leads to an error uncertainty of about $\sim 2.75 \times 10^{15}$ molec/cm² on the tropospheric column from direct sun data. It should be noted that this approach leads to retrieval of total tropospheric column from the direct sun, while the tropospheric column from MAX-DOAS represents mainly the boundary layer.

### 25 4. Comparison method

For the comparison, GOME-2A and OMI data were extracted within a radius of 50 km around the 36 stations listed in Table 2 and Table 3 with only pixels having a cloud radiance fraction <50% and an AMFratio (AMFtropo/AMFgeom) > 0.2 (Boersma et al., 2018) being selected. In the case of OMI, pixels affected by the row anomaly were filtered out (Boersma et al., 2018). The closest pixels and the mean value within the extraction

radius were calculated for each day. To reduce the differences in spatial resolution of the satellite measurements (GOME-2A: 40x80km², OMI: 13x24km² at best) compared to the ground-based sensitivity (horizontal length of the probed air mass up to ~20 km), the largest pixels from each instrument dataset were removed: only pixels with an across-track width smaller than 100km for GOME-2A and smaller than 40km for OMI were kept in the comparisons. Previous studies have investigated the use of stricter coincidence criteria as a way to overcome

spatial resolution differences. E.g. Irie et al. (2008) showed differences up to 25% in satellite VCD between pixels located 5 to 50 km away from the site and only OMI pixel centered within 0.1°×0.1° of the MAX-DOAS stations were considered in the validation. Other approaches have averaged MAX-DOAS VCDs made in several azimuth directions (Brinksma et al., 2008; Celarier et al., 2008; Ortega et al, 2015) or have excluded MAX-DOAS



measurements with a relative uncertainty ≥10% (Vlemmix et al., 2010). In addition to our baseline, tests on those parameters are performed in Sect. 7.3.

5 Ground-based MAX-DOAS data were interpolated to the satellite overpass time and a verification of the presence of data within ±1h was performed in order to avoid large interpolation errors. Pandora direct sun measurements have a much higher acquisition rate (approximately 30 acquisitions/hour compared to typically 1 to 4 MAX-DOAS measurements) with sometimes strong $NO_2$ variations not perfectly removed with the data filtering, so Pandora measurements within 1 hour (±30min) of the satellite overpass time were averaged. On this basis, daily comparisons were performed at each station, and corresponding monthly averages were also calculated.

As an example, Fig. 2 shows the results of our analysis for the Xianghe MAX-DOAS site. Pollution episodes are well captured by both GOME-2A and OMI as well as seasonal variations characterized by high $NO_2$ VCDs in winter and low values in summer. Quantitatively, the comparison of the whole time-series is good, with correlation coefficients R of 0.88 and 0.9 and linear regression slopes of about 0.71 and 0.78, for the monthly GOME-2A and

15 OMI data respectively. VCDtropo differences (SAT-GB in x$10^{15}$ molec/cm²) and percent relative difference (100*(SAT-GB))/GB in %) were calculated for each site. For Xianghe the median bias is of about -2.3 x$10^{15}$ molec/cm² (-9%) and 0.13 x$10^{15}$ molec/cm² (1%) for GOME-2A and OMI data respectively. Values for each site are reported in Table S1 for GOME-2A and OMI, with daily and monthly statistics for correlation coefficient R, slope S and intercept I of a linear regression and mean and median monthly absolute and relative biases. Depending

20 on the length of the ground-based time-series, the number of daily comparison points can vary significantly, from at least 3 months of data to several years of continuous measurements.

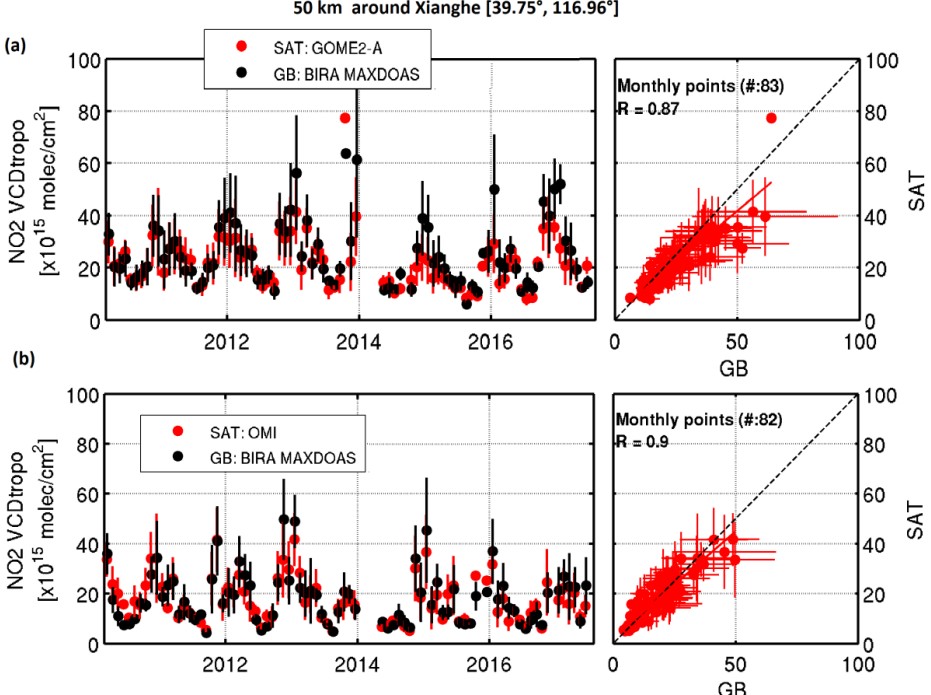

**Figure 2**: Comparison of monthly mean tropospheric $NO_2$ VCDs at the Xianghe station for (a) GOME-2A GDP 4.8 data and (b) OMI DOMINO v2.0 versus MAX-DOAS data, over the period March 2010 to July 2018. Correlation coefficients R are given as insert on the scatter plots on the right column. The variability (standard deviation of the monthly mean) is given as error bars for both datasets.

## 5. Results

### 5.1 Overview of the ground-based datasets

Figure 3 presents an overview of the tropospheric and stratospheric $NO_2$ columns measured at each station, as obtained from the satellite-to-ground based coincidences. The tropospheric columns correspond to the ground-based data as selected in Sect.4 (including, for the direct sun case, the subtraction of the satellite-estimated stratospheric content, see Sect. 3), while the stratospheric columns are the satellite estimations. As can be seen from the box and whisker plot, the tropospheric content varies strongly among the stations, the observed median columns ranging from 1 $x10^{15}$ molec/cm² in rural places (Hohenpeisseberg, Reunion, Cape Hedo, Mauna Loa, Izaña) to about 30 to 40 $x10^{15}$ molec/cm² in highly polluted sites (Beijing, Seoul, Beijing-CMA). As can also be seen, tropospheric columns selected at GOME-2A overpass times (i.e. in the morning) are usually larger than those selected at OMI overpass time (13:30±0:90), which is explained by lower OH levels and somewhat higher NOx emissions leading to slower $NO_2$ chemical loss in mid-morning (09:30 hrs) compared to noon (13:30 hrs) (Boersma et al., 2008; Kim et al., 2009). Note that the median tropospheric column is negative at the mountain top stations of Izaña and Mauna Loa. This is either caused by a slight underestimation of the Pandora total columns or a slight





overestimation of the stratospheric columns derived from satellite. This discrepancy is under investigation and will be the subject of a future study.

Due to different deployment strategies, the direct sun instruments (mostly Pandora's type) tend to be located closer
to strong $NO_2$ emission sources than MAX-DOAS instruments, that sample both polluted and background sites. The MAX-DOAS ensemble of stations has more stations in the 2 to 20 $x10^{15}$ range (about 18 MAX-DOAS stations and 10 direct sun stations). Moreover, being able to also measure under cloudy conditions, MAX-DOAS sites tend to sample the full variability of the $NO_2$ field at measurement sites, while direct sun data preferentially sample clear-sky conditions. As a result, MAX-DOAS sites tend to display a larger variability, as can be judged from the
larger boxes (25 to 75 percentile) and lines (9 to 91 percentile) in the box and whisker plots of Figure 3a.

Figure 3b presents the stratospheric columns derived from the two satellites. Values typically range between 2 $x10^{15}$ and 3.5 $x10^{15}$ molec/cm². The difference of about 0.6 (up to 1) $x10^{15}$ molec/cm² between the GOME-2A and OMI data is consistent with the known diurnal variation of the stratospheric $NO_2$, which results from the $NO/NO_2$ equilibrium and the progressive photo-dissociation of $N_2O_5$ during the day (Dirksen et al., 2011; Belmonte-Rivas
et al., 2014 ; van Geffen et al., 2015). Minimum values of the stratospheric column are obtained over the equatorial sites (Nairobi, Bujumbura and Mauna Loa).

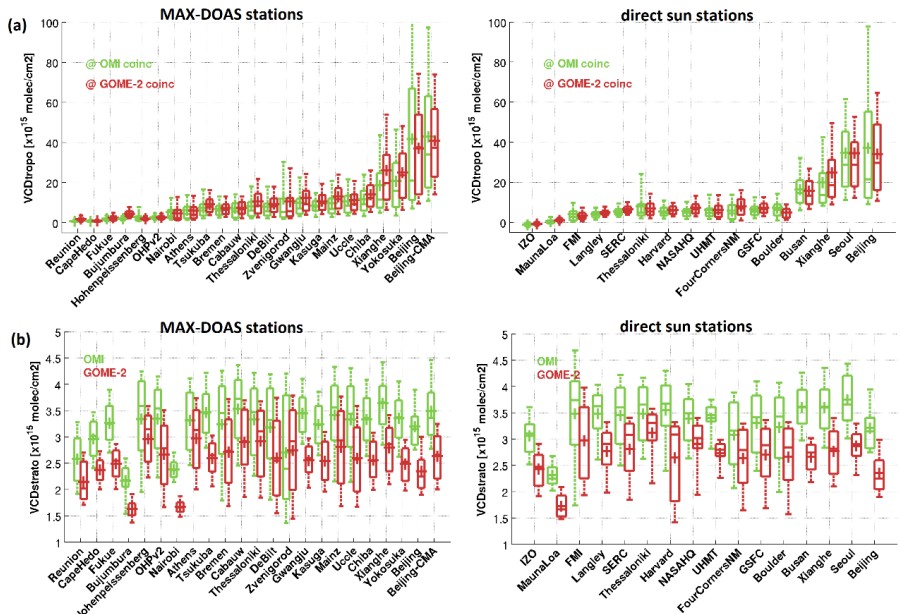

**Figure 3**: $NO_2$ columns at the various ground-based stations (MAX-DOAS on the left panels and direct sun on the right panels): (a) box and whisker plot of the ground-based tropospheric $NO_2$ columns (obtained by subtracting the satellite VCDstrato in the
case of direct sun data), (b) box and whisker plot of the stratospheric $NO_2$ content derived from satellite instruments. OMI data in green, GOME-2A data in dark red. The box and whisker plots are defined as follows: crosses for the mean values, horizontal lines for the median, boxes for the 25[th] and 75[th] percentile and vertical lines for the 9[th] and 91[th] percentile. Stations are ordered by increasing values of the VCDtropo columns.





The validity of the tropospheric estimation approach applied to the direct sun data (see Section 3.2 and Eq. 2) was verified at stations where both MAX-DOAS and direct sun measurements are performed. This is the case for 3 sites: Beijing, Xianghe and Thessaloniki. Combining these 3 data sets, Figure 4 displays a scatter plot of the tropospheric $NO_2$ columns measured by both techniques. Results are shown separately for GOME-2A and OMI

overpass times. In both cases, a high level of correlation is obtained (linear correlation coefficient > 0.94). The corresponding linear regression slopes are 1.17±0.01 and 1.12±0.01 for OMI and GOME-2 overpasses respectively, with intercepts of -5 x$10^{15}$ molec/cm² and -2.8 x$10^{15}$ molec/cm². These results suggest that MAX-DOAS and direct sun data show a small relative bias of about 10-15 percent. Part of this bias, which could change depending on pollution levels, may arise from the satellite-based stratospheric correction applied to direct sun data.

However, it should be noted that MAX-DOAS and direct sun measurements are not synchronized, with typical differences in measuring time of about half an hour for these stations. The $NO_2$ variability (which can be large in polluted sites) therefore probably contributes to the observed scatter and apparent bias. Furthermore, MAX-DOAS and direct sun instruments observe different air masses, which might lead to differences in the presence of horizontally inhomogeneous air masses.

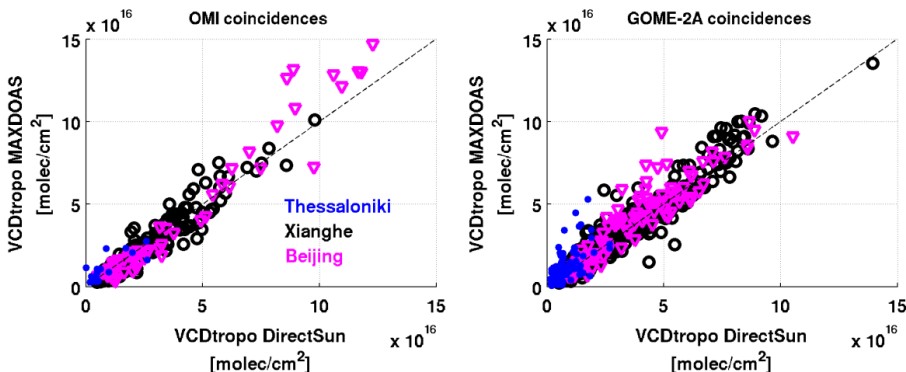

**Figure 4**: MAX-DOAS and direct sun tropospheric $NO_2$ columns in Thessaloniki, Xianghe and Beijing. At these sites, ground-based measurements are performed in both geometries.

Another approach to verify the consistency of the ground-based dataset is to investigate the coherence between measurements at sites that are geographically close to each other. For example, NASA-HQ and GSFC are very close to each other, but measurements were performed by different Pandora instruments and during different time-periods. Their median VCDtropo differences for the overlapping days are about 1.5 and 7.4 x$10^{14}$ molec/cm² at the OMI and GOME-2 overpasses respectively, in line with the expected uncertainty/variability of these ground-

based data. Beijing and Beijing-CMA sites are interesting to compare since both are located inside the city, at a mutual distance of about 6 km. The first instrument has been measuring on the roof of the Institute of Atmospheric Physics (IAP) (Clémer et al., 2010), the second at the China Meteorological Administration (Ma et al., 2013). Both instruments have already been compared in Hendrick et al. (2014) showing good agreement (differences of about -2% in winter and 3 to 4% for the rest of the period). When comparing their columns for the satellite's colocations,



they present differences of about 1 and 6 x10$^{15}$ molec/cm² at OMI and GOME-2A overpass times, respectively (5 to 15%). Another example is Chiba and Yokosuka. Both of these sites are situated on the urban Tokyo bay but at about 53km distance from each other. Their median differences from OMI and GOME-2 are 5.7 and 14.2 x10$^{15}$ molec/cm² respectively (69 to 82%).

### 5.2 Comparison of ground-based and satellite datasets

The comparison methodology illustrated in Fig. 2 has been extended to the 23 MAX-DOAS and 16 direct sun stations gathered in this study. As expected, results show a clear dependence on the location of the comparison site. The best agreement is obtained in background/remote conditions while comparisons are more challenging
close to the sources, where the NO$_2$ field is more heterogeneous (Chen et al., 2009; Irie et al., 2012; Ma et al., 2013; Pinardi et al., 2014). To illustrate this point, the different stations have been qualitatively classified into urban, sub-urban and background sites (see Tables 2 and 3), based on their location with respect to known pollution sources. This classification is not based on NO$_2$ levels but reflects the influence of the surrounding areas. E.g. Xianghe station is in a polluted background with high NO$_2$ levels (see Fig. 3), but it is located at a relatively large
distance from surrounding urban areas, and is thus classified as sub-urban.
Figure 5 presents monthly mean scatter plots of the GOME-2A GDP 4.8 data against ground-based measurements at the different stations. Different sites are plotted in different colors, and results are grouped separately for MAX-DOAS and direct sun data as well as for urban and background/sub-urban stations. As can be seen, satellite and ground-based data generally correlate well, with correlation coefficients ranging between 0.76 and 0.96 and linear
regression slopes between 0.35 and 0.83. For more details on the statistical analysis of the regressions, see Table 4. It is clear that smaller slopes and larger biases are found at urban locations compared to background/sub-urban ones. Note also that for the case of the comparisons against MAX-DOAS data in background/sub-urban sites, smaller biases are obtained for OMI than for GOME-2 (about -10% compared to about -44%), while in urban sites the differences among the two satellites are smaller (about -36.6% and -38%). For direct sun sites both satellite
sensors seem to behave similarly, with biases of about -20% in background/sub-urban cases and about -25% in urban cases (Table 4).



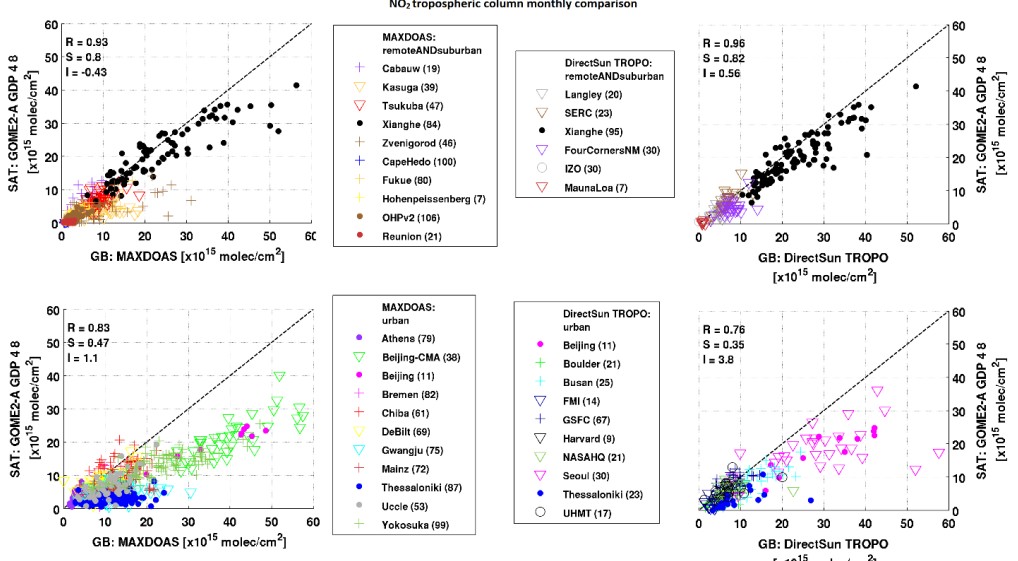

**Figure 5**: Scatter plot of GOME-2A GDP 4.8 NO₂ tropospheric columns with respect to MAX-DOAS instruments (left panels) and direct sun instruments (right panels). Upper plots display background and sub-urban stations, while urban stations are represented in the lower plots. Linear regression values are given as insert for each case (correlation coefficient R, slope S and intercept I) and the number of months for each station is given in brackets in the legends. Pixel selection: GOME-2A pixel size < 100km (i.e. removing back scans).

**Table 4**: Statistics of the monthly median comparisons per station type for the satellite versus ground-based comparisons. Linear regression slope S and intercept I are presented.

| | R | S | I [x10¹⁵ molec/cm²] | Bias (SAT-GB) [x10¹⁵ molec/cm²] | Bias % |
|---|---|---|---|---|---|
| **MAX-DOAS comparisons** | | | | | |
| **Sub-urban & remote** | | | | | |
| GOME-2A | 0.93 | 0.8 | -0.43 | -4.9 | -44% |
| OMI | 0.77 | 0.58 | 1.47 | -0.2 | -9.6% |
| **Urban** | | | | | |
| GOME-2A | 0.83 | 0.47 | 1.1 | -0.99 | -38% |
| OMI | 0.84 | 0.55 | 1.16 | -3.3 | -36.6% |
| **With dilution correction:** | | | | | |
| **Sub-urban & remote** | | | | | |
| GOME-2A | 0.93 | 0.77 | -0.24 | -1 | -36% |
| OMI | 0.79 | 0.57 | 1.49 | -0.26 | -12% |
| **Urban** | | | | | |
| GOME-2A | 0.83 | 0.69 | 0.55 | -3.2 | -27% |



| | | | | | |
|---|---|---|---|---|---|
| **OMI** | 0.84 | 0.83 | 0.31 | -1.44 | -17% |
| **direct sun tropospheric comparisons** | | | | | |
| **Sub-urban & remote** | | | | | |
| **GOME-2A** | 0.96 | 0.82 | 0.56 | -1.15 | -19.5% |
| **OMI** | 0.95 | 0.80 | 0.81 | -0.96 | -20.2% |
| **Urban** | | | | | |
| **GOME-2A** | 0.76 | 0.35 | 3.81 | -2.3 | -27% |
| **OMI** | 0.69 | 0.42 | 3.19 | -1.75 | -25% |
| **With dilution correction:** | | | | | |
| **Sub-urban & remote** | | | | | |
| **GOME-2A** | 0.95 | 0.75 | 1.61 | -0.87 | -17.5% |
| **OMI** | 0.93 | 0.73 | 1.84 | -0.65 | -20.2% |
| **Urban** | | | | | |
| **GOME-2A** | 0.77 | 0.59 | 4.64 | -0.08 | -0.95% |
| **OMI** | 0.71 | 0.70 | 3.7 | 0.33 | 3.8% |

The median relative biases (SAT-GB)/GB at each site are presented as a color-coded map in Figure 6. Satellite data display a negative bias against ground-based reference data at all stations, except Reunion Island and UHMT-Houston, which are both coastal sites, highly heterogeneous in nature (Tzortziou et al., 2014; 2015; 2018; Loughner et al. 2014; Martins et al., 2016). Negative biases of about -80% are observed in Bujumbura and Nairobi, which can be related to the small $NO_2$ signal and the localized nature of the sources at these sites, combined with a complex orography (Gielen et al., 2017; Compernolle et al., 2020). Systematic uncertainties in the estimation of the stratospheric column in satellite datasets could also contribute to the observed underestimation, considering the overall small tropospheric $NO_2$ signals at these locations. E.g. Valks et al. (2011) have shown that small-scale variations visible in the IFS-MOZART stratospheric $NO_2$ field could not be captured by the GOME-2A stratosphere-troposphere separation algorithm, due to limitations of the spatial filtering approach. In particular this might be the case at the Izaña and Mauna Loa stations (see Fig. 3a), where the satellite stratospheric column is found to exceed the total column $NO_2$ derived from ground-based direct sun measurements. Finally, issues related to the use of inadequate ancillary datasets might also affect the accuracy of the satellite $NO_2$ columns. This can be due to the coarse spatial resolution of models used as a priori (from 1.875° to 3° here, see Table 1) or their temporal sampling (monthly values from 1997 or daily profiles, see Table 1), leading to unrealistic representation of the sources and errors on the AMF calculation of up to 50% (Heckel et al., 2011; Lin et a. 2014; Kuhlman et al., 2015, Laughner et al., 2016; 2019; Judd et al., 2019). Also Liu et al. (2019c) showed that known uncertainties in albedo climatologies result in $NO_2$ column uncertainties of 3-6%, while errors in model input are responsible for up to 20% of error on the retrieved $NO_2$ columns.

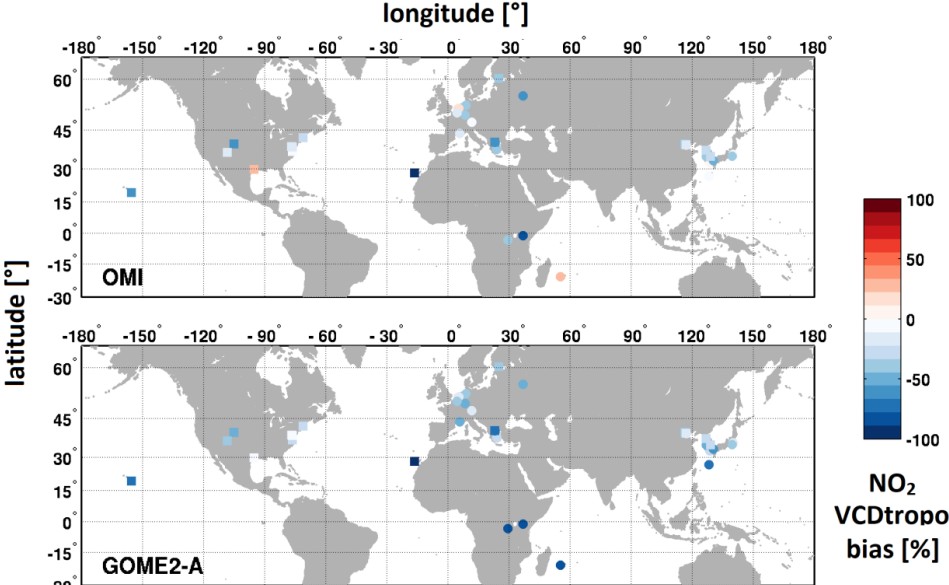

**Figure 6**: Daily median relative bias at each station for OMI DOMINO v2 and GOME-2A GDP tropospheric NO$_2$ columns. MAX-DOAS stations are represented with circles, and direct sun stations with squares.

Looking at the details of the comparison results at each station (Fig. 6 and values in Table S1), we find that GOME-2A and OMI present a similar behavior at a large number of stations. Biases, however, tend to be slightly larger for GOME-2A. E.g., in the Beijing megacity, the median monthly mean bias is -35% for OMI and -43% for GOME-2A when considering direct sun cases, -46% and -49% for the Beijing MAX-DOAS case, and -35% and -48% for Beijing-CMA MAX-DOAS case. In Xianghe, which is a sub-urban site, the biases are smaller (1% and -9% for MAX-DOAS), as expected. Table S1 provides a complete overview of the monthly bias results obtained when comparing OMI and GOME-2A to MAX-DOAS and direct sun instruments. Aside from the stations showing coherent validation results for OMI and GOME-2A (about 10 out of 16 direct sun sites and 10 out of 23 MAX-DOAS sites with differences in the satellite-to-ground validation results bias of less than 15%), others are characterized by much larger differences, especially in remote sites such as OHP, Reunion, Cape Hedo, Fukue, Tsukuba and Bujumbura. A few mountain-top or high altitude sites present very large relative biases such as Nairobi (-88%), Mauna Loa (-60% to -120%) and Izaña (-190 to 190%). At Reunion and Bujumbura, only GOME-2A results display large biases (-79% compared to 14% for Reunion, and -84% compared to -31% for Bujumbura). Significant differences between ground-based MAX-DOAS and both OMI QA4ECV and OMI NASA were also reported by Compernolle et al. (2020) in OHP, Bujumbura, Nairobi and Mainz.

When considering the results as a whole, the most prominent feature is the systematic underestimation of ground-based data by both satellite datasets for most of the sites. This underestimation is mostly prominent at urban sites close to the sources, but it is also found at background/sub-urban sites and cannot be fully explained by the satellite



uncertainties (see section 2). The differences observed between OMI and GOME-2A can be related to instrumental characteristics (e.g. differences in pixel size) but also to details of the applied retrieval methods (see Table 1 and Sect. 2). Several studies have discussed in detail the impact of algorithmic differences on the $NO_2$ column uncertainty, which can reach 42%, mainly due to tropospheric AMF uncertainties (Lorente et al., 2017). The

underestimation of the $NO_2$ satellite products identified here at a large number of stations, confirms what was obtained in previous validation exercises using fewer sites and different satellite products (Celarier et al., 2008; Brinksma et al., 2008; Vlemmix et al., 2010; Irie et al., 2008; Lin et al., 2014; Halla et al., 2011; Irie et al. 2012; Shaiganfar et al., 2011; Ma et al., 2013; Kanaya et al., 2014; Wang et al. 2017b; Mendolia et al. 2013; Tzortziou et al., 2014; Lamsal et al., 2014; Drosoglou et al., 2017; Herman et al., 2019; Judd et al., 2019; Compernolle et al.

2020). These studies generally reported small negative or positive biases over rural (unpolluted) measurement sites and stronger (systematic) negative biases over urban polluted sites.

One way to understand these results is to consider the impact of the spatial resolution of the satellite measurements. For the case of rural sites, coincident satellite pixels can include areas with higher $NO_2$ columns leading to positive

biases in the comparisons. In contrast at urban locations characterized by strong $NO_2$ sources, coincident pixels generally tend to include surrounding (sub-urban) areas. This effect is especially significant for satellite instruments measuring at coarse spatial resolution, such as GOME-2A. It can be attenuated in validation studies making use of long time-periods and many stations, however large localized $NO_2$ concentrations will always tend to be underestimated. This is particularly true for satellite instruments characterized by horizontal resolution much

coarser than the size of typical urban agglomerations (see Table 1). Note that the effect can be somewhat mitigated in the case of satellite retrievals using a-priori profiles specified at high temporal and spatial resolution (Huijnen et al., 2010; Russell et al. 2011, Heckel et al., 2011; Lin et al., 2014; McLinden et al. 2014; Kuhlmann et al., 2015; Laughner et al. 2019; Goldberg et al., 2017; 2019). In the next section, we present an attempt to quantify the smearing effect around urban sites and use it to study its impact on validation results.

## 6. Horizontal dilution effects

In order to investigate the horizontal variability of the $NO_2$ field at the 36 different stations, one full year (2005) of OMI $NO_2$ QA4ECV dataset v1.1 (Boersma et al., 2018) was extracted to map the average $NO_2$ column distribution at a grid of 0.025°x0.025° in latitude-longitude. Such highly-resolved gridded maps were obtained

using a realistic representation of the OMI point spread function allowing to subsample the native OMI pixels (Sihler et al. 2017). Only the smallest OMI pixels (rows 11 to 49) were retained for this analysis. Corresponding high resolution grids were used to quantify the systematic change in tropospheric $NO_2$ between the position of the satellite pixels and the location of the stations, what we call hereafter the dilution effect. The approach used here is an extension of a similar method introduced by Chen et al. (2009) and Ma et al. (2013) based on high resolution

city night lights maps used as a proxy for $NO_2$ sources. Judd et al. (2019) also accurately quantified this effect in the New-York area using airborne $NO_2$ mapping data from the GeoTASO instrument. In our approach, the variation of the tropospheric $NO_2$ VCD is sampled in concentric circles of different radii around each of the stations. Figure 7 illustrates the method for the Beijing (urban, Fig 7a) and Xianghe (sub-urban, Fig; 7c) sites, which both present strongly inhomogeneous $NO_2$ fields. Figure 7b and 7d shows the $NO_2$ VCD variation in



concentric circles around the stations. In Beijing, the ground-based instrument is located close to the urban $NO_2$ hotspot, so that the $NO_2$ level decreases rapidly outwards. In contrast, a different behavior is found at the Xianghe station, which is located at about 60 km to the East of the Beijing city center. In this case, due to the influence of the surrounding emission sources, the mean $NO_2$ column tends to slightly increase when moving away from the

site in the direction of Beijing. For background sites, one expects the $NO_2$ content to remain roughly constant around the station value. Horizontal variability effects have been documented in previous studies dealing with ozone and water vapor (Lambert et al., 2012, Verhoelst et al., 2015), as well as with tropospheric $NO_2$ (Irie et al., 2012; Duncan et al., 2016 and Boersma et al., 2018), mostly to illustrate the impact of collocation mismatch errors on validation results. In our study, we propose a correction method applied to satellite data, which aims at reducing

the impact of the smearing effect on comparisons.

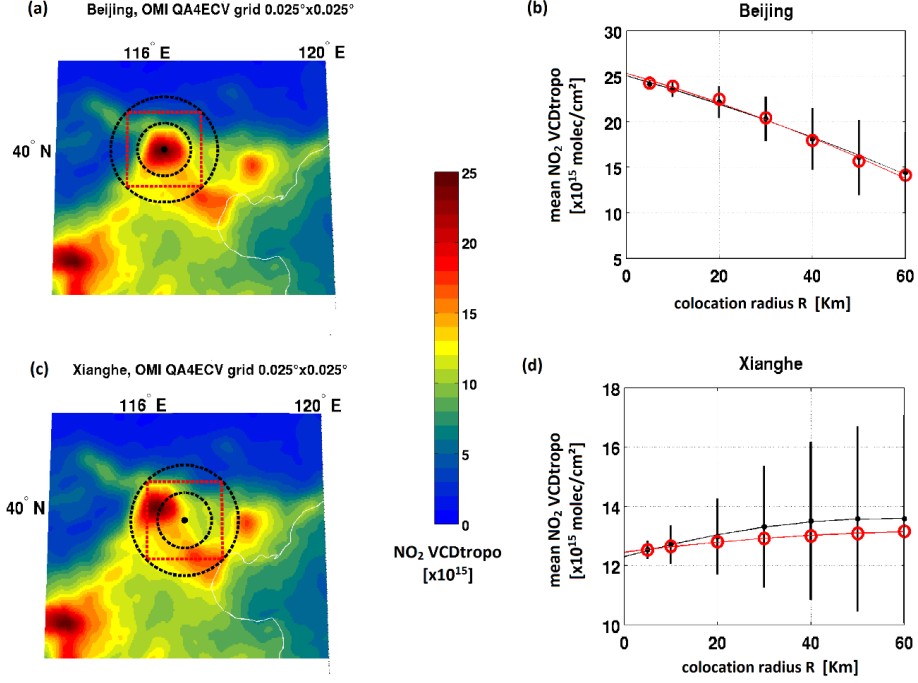

**Figure 7**: Dilution effect illustration on typical urban (Beijing, upper row) and sub-urban (Xianghe, lower row) case. Left panels represent the 2005-yearly mean tropospheric $NO_2$ gridded from OMI QA4ECV data at the resolution of 0.025° latitude x 0.025° longitude. The black dot indicates the station location, the 2 circles denote 50 and 100 km radii around the station and

the red box represents the extension of the GOME-2 pixels when the center of the pixels is within the 50km radius. Right panels display the mean (black) and median (red) $NO_2$ values at increasing colocation radii (expressed in km), with the variability (one standard deviation) given as an error bar around the mean.

### 6.1 Dilution correction method

Similarly to the studies of Chen et al. (2009) and Ma et al. (2013), a correction factor is calculated to quantify the change in $NO_2$ between the ground-based site and the satellite pixel location. In our approach, the dilution factor



($F_{dil}$) is obtained from the OMI gridded files by taking the ratio between the average (mean or median) $NO_2$ VCD at increasing distances from the site and the VCD value at the site. A second order polynomial is then fitted to these ratio values as illustrated in Fig. 7 (panels (b) and (d)). Accordingly, $F_{dil}$ is calculated using the following equation, where R represents the distance from the site:

$$F_{dil}(R) = NO2\_VCD(R)/NO2\_VCD(0) \tag{3}$$

In practice, $F_{dil}$ is calculated as the median values of the gridded $NO_2$ field for values of R from zero to 50km. For sites showing a negative slope in the dilution factor (i.e. a clear dilution effect, see figures S3 and S6 to S30 in
supplement) a dilution correction (DC) is applied to the satellite data according to:

$$VCDsat\_DC = VCDsat/F_{dil}(R) \tag{4}$$

This correction is applied to individual satellite measurements according to their respective distances. Typically,
it is applied to large urban sites, stations isolated on small islands such as Reunion Island (Fig. S18), Izaña (Fig. S15) and Mauna Loa (Fig. S27), stations close to a large power plant such as FourCorners (Fig. S11), and generally speaking sites characterized by a $NO_2$ hotspot surrounded by a clean area. The stations where a dilution correction was applied are (from North to South): Helsinki FMI, Bremen, De Bilt, Uccle, Mainz, Harvard, Thessaloniki, Boulder, Beijing, Beijing-CMA, NASA-HQ (Head Quarters), GSFC, Athens, Seoul, Yokosuka, Langley,
FourCorners (New Mexico), Chiba, Busan, Gwangju, Kasuga, UHMT, Izaña (IZO), Mauna Loa and Reunion Island (LePort station). This ensemble is referred to as UIPP (Urban, Island and Power Plant) in the rest of the paper.

**6.2 Impact of the dilution correction**

The regression analyses discussed in section 5 have been repeated after application of the dilution correction to the satellite data sets. Results obtained with and without the correction are detailed in Table 4 for MAX-DOAS and direct sun data sets. We also distinguish between urban and background/sub-urban cases. Since the background sites are generally not (or weakly) influenced by dilution effects, results do not change much for this category, as expected. In contrast, when considering urban sites, statistical parameters are significantly modified. Regression
slopes tend to be larger and closer to unity while biases are systematically reduced by about 10 to 20%.

The improvement brought by the dilution correction is further illustrated in Fig. 8, where the slopes of the linear regressions from daily scatter plots are presented for each station separately with and without dilution correction. In order to limit the impact of outliers (especially the large columns that strongly affect the regression analysis),
daily comparison points are filtered for values larger than the 75th percentile of the ground-based values of each station. This selection excludes large local values that cannot be captured by satellite measurements and allows for a more robust statistical regression analysis. In each panel, the case denoted "all" corresponds to a combined analysis including the data from all stations together. This is different than the average slope of the stations slopes,





as the different sites have varying number of points. After application of the dilution correction, regression slopes improve (and come closer to unity) for all cases except De Bilt. However, for some sites, there seems to be an over-correction effect (Athens/GOME-2, UMHT/GOME-2, Beijing (both sites)/OMI and Reunion/OMI), while negative slope are obtained at a few other sites (e.g. Mauna Loa/GOME-2 and Reunion/GOME-2). As already
discussed in Section 5.1, for direct sun stations this could be related to issues with the determination of stratospheric columns in the satellite algorithm. UHMT is a peculiar site, where several studies performed during the DISCOVER-AQ 2013 Texas campaign (Nowlan et al., 2018; Choi et al., 2019) suggested that those Pandora $NO_2$ measurements tend to be too low. Finally, some sites (e.g. Nairobi, Bujumbura, Thessaloniki, Izaña) display very small slopes probably due to the fact that these sites are characterized by very local sources or by non-
symmetric $NO_2$ distributions. This is clearly the case for isolated islands where the $NO_2$ can be locally trapped due to orography (see figures S19, S22, S24 in supplement).

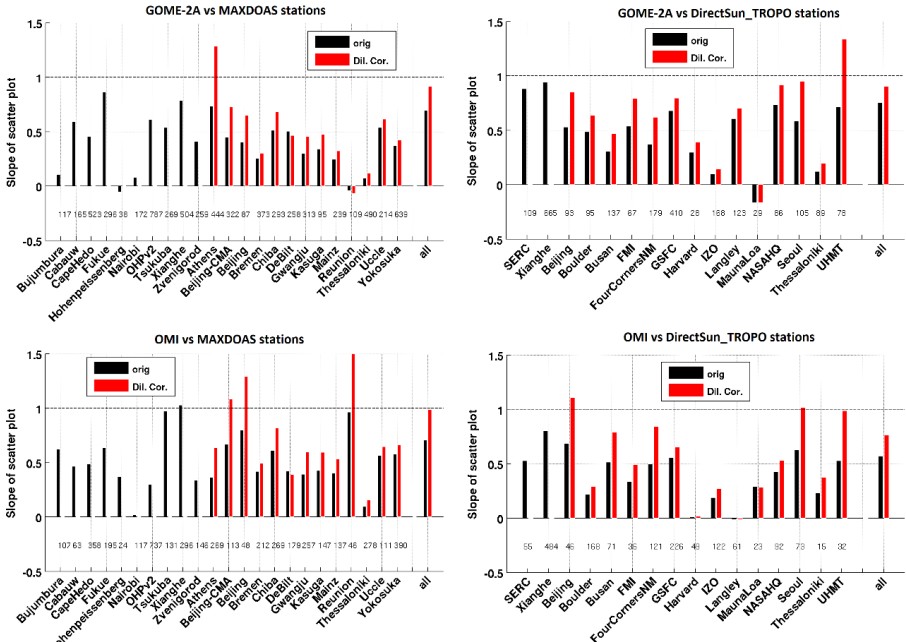

**Figure 8**: Bar plot of the daily regression slopes at each station for the original (black bars) and the dilution corrected data (red bar, for the UIPP stations). In order to reduce the weight of large columns on the regression line and to remove local effects,
data are filtered to keep only points smaller than percentile 75. Left column: MAX-DOAS stations, right column: direct sun stations. First row: GOME-2A GDP, second row: OMI DOMINO v2.0.

Figure 9 displays monthly scatter plots of satellite and ground-based data for all the UIPP stations, i.e. those at which a dilution correction was applied. Data points corresponding to values larger than the 75th percentile are
represented as grey points. The two upper plots show results without correction for MAX-DOAS (left) and direct sun (right) data sets, while corrected data are represented similarly in the lower plots. Again, the impact of the dilution correction is clearly apparent. The regression slope increases from 0.52 to 0.76 for MAX-DOAS and from





0.67 to 1.1 for direct sun data. The impact of excluding the largest columns from the regression analysis can be judged by comparing the grey and black lines, respectively obtained without and with filtering. As can be seen direct sun data are more affected by this filtering (slope increase from 0.38 to 0.67) than MAX-DOAS ones (slope increase from 0.49 to 0.52). This is likely related to the fact that, as already mentioned, direct sun instruments (of

Pandora type) tend to be located closer to strong $NO_2$ emission sources than MAX-DOAS instruments. Table 5 lists the statistical parameters from regression analyses performed with and without the dilution correction for all the UIPP stations and the different satellite products. Generally speaking, validation results obtained using both MAX-DOAS and direct sun systems appear to be consistent, although direct sun observations tend to agree slightly better with the satellite data. In the case of direct sun data, however, we note that the dilution correction tends to

over-correct satellite measurements (see also Fig. 9). It is also interesting to note in Table 5 that the intercepts are always positive, which could point to a systematic additive bias, possibly coming from an under-estimation of the stratospheric (slant) columns. A bias of about -0.2 x1015 molec/cm² has been reported by Compernoelle et al. (2020) when comparing the OMI QA4ECV assimilated stratospheric columns (based on an approach similar to the one used in the OMI DOMINO algorithm) to ground-based zenith-sky data. This bias was reduced to about -

0.01 x1015 molec/cm² when using the STREAM (Beirle et al., 2016) approach.

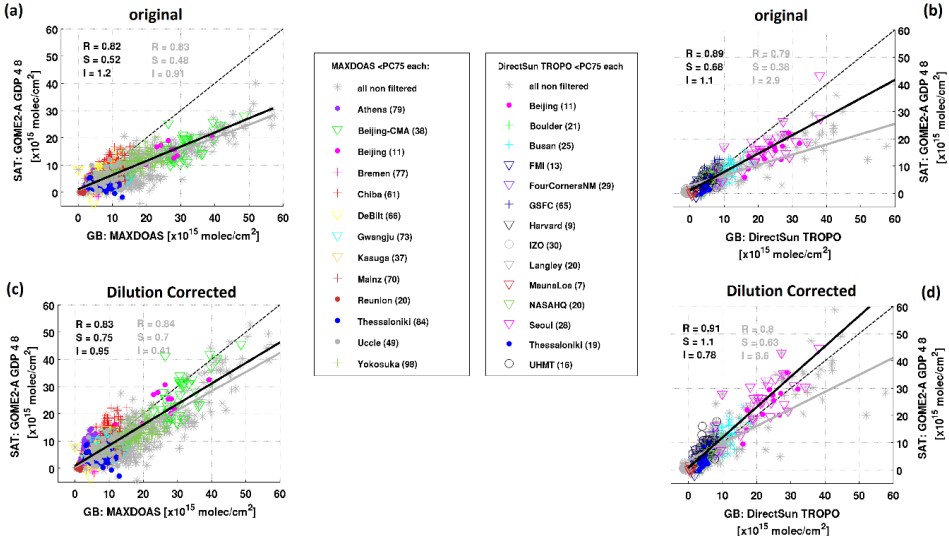

**Figure 9**: Scatter plot of monthly mean GOME-2A GDP 4.8 $NO_2$ columns with versus UIPP ground-based stations (MAX-DOAS instruments on the left panels and direct sun instruments on the right panels). The upper panels present the original comparisons and the lower panels those after applying the dilution correction. Calculations of the monthly mean values are

performed after removal of the daily ground-based points larger than percentile 75 of each station dataset. The monthly means without the filtering are presented in grey to illustrate the impact and the number of remaining months for each station is given in brackets in the legends. Linear regression values are shown on each plot.

**Table 5**: Statistics of the monthly median comparisons of ground-based with satellite data for UIPP ensembles, before and

after the PC75 filtering and the Dilution Correction are applied.

| | R | S | I | Bias | Bias |
|---|---|---|---|---|---|



| | | [x10$^{15}$ molec/cm²] | (SAT-GB) [x10$^{15}$ molec/cm²] | % |
|---|---|---|---|---|
| **MAX-DOAS comparisons** | | | | |
| **All original** | | | | |
| **GOME-2A** | 0.83 | 0.48 | 0.9 | -4.77 | -44.5% |
| **OMI** | 0.85 | 0.56 | 1.02 | -3.3 | -36.8% |
| **Original filtered** | | | | |
| **GOME-2A** | 0.81 | 0.52 | 1.16 | -2.8 | -37.3% |
| **OMI** | 0.8 | 0.65 | 0.97 | -1.63 | -26% |
| **All With dilution correction:** | | | | |
| **GOME-2A** | 0.84 | 0.69 | 0.4 | -3 | -28.5% |
| **OMI** | 0.85 | 0.83 | 0.26 | -1.45 | -17.3% |
| **filtered With dilution correction:** | | | | |
| **GOME-2A** | 0.83 | 0.76 | 0.94 | -1.37 | -18.4% |
| **OMI** | 0.83 | 0.99 | 0.5 | 0.08 | 1.8% |
| **direct sun tropospheric comparisons** | | | | |
| **All original** | | | | |
| **GOME-2A** | 0.79 | 0.38 | 2.9 | -1.63 | -29.4% |
| **OMI** | 0.74 | 0.44 | 2.65 | -1.11 | -28.3% |
| **Original filtered** | | | | |
| **GOME-2A** | 0.89 | 0.67 | 1.13 | -5.27e14 | -22% |
| **OMI** | 0.82 | 0.67 | 1.45 | -8.9e12 | -16.4% |
| **All With dilution correction:** | | | | |
| **GOME-2A** | 0.80 | 0.63 | 3.62 | 2.08e14 | -5.7% |
| **OMI** | 0.74 | 0.72 | 3.22 | 7.27e14 | 2.36% |
| **Filtered With dilution correction:** | | | | |
| **GOME-2A** | 0.91 | 1.11 | 0.78 | 1.18e15 | 11.1% |
| **OMI** | 0.83 | 1.11 | 1.45 | 1.37 | 12.8% |

Considering all the stations together, Figure 10 presents an overview of the differences between satellite and ground-based data sets, for the original comparisons (in black) and after dilution correction (in red). We make the distinction between two different approaches for the selection of the coincident pixels: closest cloud free (cloud radiance fraction<50%) pixel and mean value of all cloud free pixels within a radius of 50 km. Results are also given separately for MAX-DOAS sites (upper plot) and direct sun sites (lower plot).

As can be seen, the overall agreement between satellite and ground-based data sets is better for OMI comparisons and, after dilution correction, it is slightly better for direct sun than for MAX-DOAS sites. Again, this is likely related to the fact that direct sun instruments (of Pandora type) tend to be located closer to strong NO$_2$ emission sources. Moreover, as also discussed previously, MAX-DOAS sites report measurements under a larger variability of conditions (both clear-sky and cloudy), leading to an increased spread of the comparisons. Generally speaking

the dilution correction pushes biases closer to zero and often reduces the spread of the differences. The best results are obtained with OMI, when comparing direct sun tropospheric columns to the closest pixel of the satellite. In this case, the median bias of -1.16 x10$^{15}$ molec/cm² obtained is reduced to -0.23 x10$^{15}$ molec/cm² after application of the dilution correction. A similar improvement is found for the MAX-DOAS comparisons, from -0.95 to -0.47

5  x10$^{15}$ molec/cm²). We find that the selection of the daily closest pixel leads to smaller biases and spreads and a better agreement between median and mean values for both OMI and GOME-2A comparisons. Therefore, in the rest of the study, comparison results are exclusively based on coincidences determined using daily closest pixels.

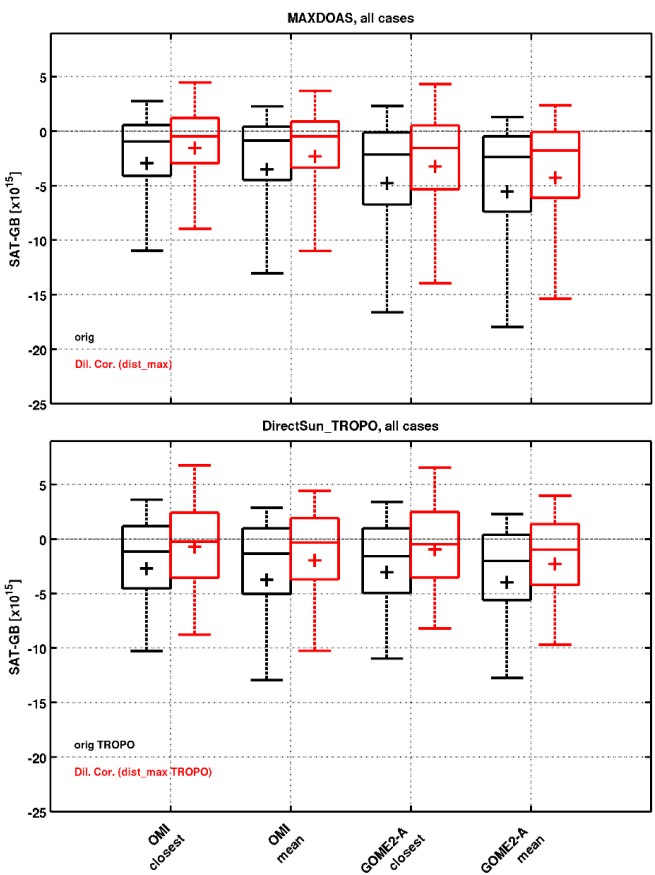

**Figure 10**: Box and whisker plot of the daily biases for all the stations with (red) and without (black) dilution correction (see

10  Sect. 6.1). First row: ensemble of MAX-DOAS stations, second row: ensemble of direct sun stations. For each row, several cases are shown: closest pixel and mean value within the 50 km radius for OMI DOMINO v2.0 and GOME-2 GDP 4.8. The box and whisker plots are defined as follow: crosses for the mean values, horizontal lines for the median, boxes for the 25$^{th}$ and 75$^{th}$ percentile and vertical lines for the 9$^{th}$ and 91$^{th}$ percentile.

15  Although the dilution correction improves the agreement between the ground-based and satellite measurements, significant negative biases persist at some of the validation sites (see Fig. 8). This could be related to satellite





retrieval issues but also to shortcomings in our correction approach, which relies on average NO₂ fields derived using one year (2005) of OMI data. These average fields are not necessarily representative of the actual day-to-day variability at all sites. This certainly contributes to the scatter of the comparisons, but should have a small (systematic) effect on regression slopes. Seasonal behavior differences, not taken into account here, could also play a role. Moreover the OMI QA4ECV dataset (Boersma et al., 2018), which has been selected as a source for estimating the correction factors, might have its own limitations. Trends in the last decades in NO₂ values worldwide (Duncan et al., 2016; Georgoulias et al., 2019) can be a limiting factor for some of the stations. Using OMI for the correction also implies that the afternoon NO₂ is representative for the morning GOME-2 overpass, which is not entirely true. Another issue is the limited spatial resolution of OMI data and of its a-priori profiles assumption. High-resolution models (Drosoglou et al. 2017) or airborne imaging DOAS measurements (Judd et al., 2019) could provide a better source of information to correct the NO₂ distributions around the stations, but such data are currently not available at the global scale.

Finally, ground-based instruments are assumed to provide point source measurements, while in reality the horizontal sensitivity area of MAX-DOAS measurements can be as large as several tens of km (Irie et al., 2011). The provision of this information for all ground-based measurements would thus be very valuable to further improve the comparison method. Note that in urban areas, the representativeness of MAX-DOAS observations for comparison with satellite data could be improved by making use of measurements in different azimuth directions (Ortega et al., 2015; Gratsea et al., 2016; Schreier et al., 2019; Dimitropoulou et al., 2020).

## 7. Seasonal variations and internal consistency of validation results

In this section, we discuss in more details the seasonal dependence of the results (Sect. 7.1) and the consistency of validation results for further changes of the pixels selection (Sect 7.2). Results for megacities such as Seoul and Beijing and for all UIPP stations are investigated in detail.

### 7.1 Seasonal dependence

Several sites submitted data for time-periods longer than one year (see Table 2 and 3 for the details), allowing to investigate the seasonal dependence of the comparisons. In Fig. 11, seasonally sorted bias values of GOME-2A and OMI against MAX-DOAS measurements are presented for six selected stations (Uccle, OHP, Beijing, Xianghe, Bujumbura and La Reunion). A dilution correction was applied to satellite data sets at three of these sites (La Reunion, Uccle and Beijing). Although comparison results are roughly consistent for all seasons, smaller biases seem to be observed in summer time at several stations of the Northern hemisphere. This might be related to the shorter lifetime of NO₂ in the warm season and the associated reduced variability of its concentration. As already discussed in Sect. 5, for Bujumbura and Reunion Island, one observes larger negative biases for GOME-2A than for OMI, despite the dilution correction applied in both sites. Note that a large under-estimation of QA4ECV OMI NO₂ VCDs was also reported by Compernolle et al. (2020) in Bujumbura. Our validation results do not point to major seasonal effects, however it is a general good practice to base validation studies on complete annual cycles in order to properly sample all observational conditions.

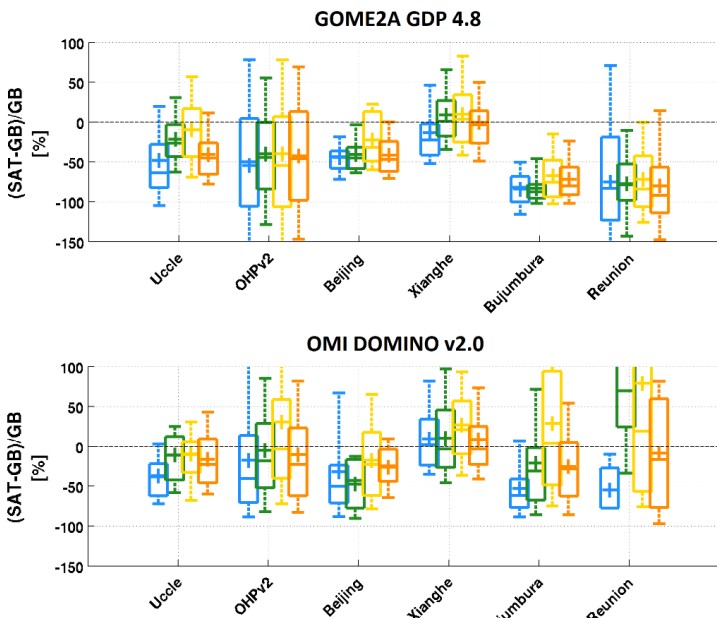

**Figure 11**: Bias (in percent) between daily tropospheric NO₂ columns from satellite (upper panel: GOME-2, lower panel: OMI) and a selection of BIRA-IASB MAX-DOAS stations, for the different seasons. A dilution correction is applied to the satellite data when relevant. The box and whisker plots are defined as in Figure 10.

**7.2 Impact of the satellite pixel selection**

To further reduce the impact of the horizontal NO₂ variability on our comparisons, an alternative approach was tested and compared to previous results. For these tests, the selection was restricted to OMI pixels covering the stations. The impact of constraining the coincidences in this way is presented in Table S1 for each station, without

10 dilution correction. The following conclusions can be drawn:

- Direct sun measurements: for 9 sites out of 16 there is a significant (more than 5%) difference between results obtained using all the pixels and only those intersecting the stations. For 6 of them, the median bias is reduced: Seoul (from -29 to -4%), Boulder (from -54 to -36%), GSFC (from -8.5 to 6.2%), Harvard (from -29 to -12%), Four Corners (from -17 to -7%) and Mauna Loa (from -120 to -60%). At three sites,

15 it is degraded: Izaña (from 190 to -210%), FMI (from -31 to 90%) and UHMT (15 to 43%).

- MAX-DOAS measurements: for 15 sites out of 23 there is a significant (more than 5%) difference between results obtained using all the pixels and only those intersecting the stations. For 10 of them, the median bias is improved: Athens (from -48 to -38%), Bremen (from -36% to -8%), Gwangju (from -44 to -34%), Kasuga (from -52 to -44%), Reunion (from 14 to 5%), Uccle (from -28 to -16%), Beijing (from

20 -39 to -24%), Thessaloniki (from -44 to -30%) and OHP (from -19 to -12%). For 5 of the sites, it is degraded: Hohenpeissenberg (from -1.3 to 17%), Tsukuba (from 3 to -6%), Bujumbura (from -31 to -46%), De Bilt (from -14% to 13%) and Fukue (-6.8 to 18%).



Thus, at most stations, using a stricter colocation criterion results in a reduction of the bias (by up to ~20%). In order to better understand the impact of changing the pixel selection criteria, additional tests were performed for two megacities characterized by extremely high $NO_2$ levels (see Fig. 3).

Figure 12 illustrates, for Beijing, Beijing-CMA, Xianghe and Seoul, the impact of making different choices on the OMI pixel size and location. For the most strict selection criterion (OMI pixels smaller than 40 km and located above the stations) we see a significant improvement of the bias and spread of the comparison in Seoul for direct sun data and only a slight improvement in the median bias for the Beijing/Beijing-CMA data. For Xianghe, the impact appears to be moderate or even negligible, as expected due to the sub-urban nature of this site (Fig. 7).

Differences in the results for the two Beijing sites are to be considered in the light of the different measurements times (Table 1) and $NO_2$ levels (Fig. 3): measurements in Beijing (median $NO_2$ of about 20 x$10^{15}$ molec/cm$^2$) were performed in 2008-2009 during the Olympic Games, while measurements at the CMA building (median of 35 x$10^{15}$ molec/cm$^2$) covered the period from 2009 to 2011. For Seoul where measurements were performed in 2012-2015 (median $NO_2$ of 35 x$10^{15}$ molec/cm$^2$), the metropolitan area extends over more than 11700 km². In this case,

as can be seen in Fig. S23, the $NO_2$ signal is in-homogeneously spread over the city and the instrument is not centered at the maximum of the satellite $NO_2$ observations. As a result, the selection of pixels in strict overpass with the site has a larger impact than for Beijing, where the MAX-DOAS instrument is located in the center of the city (Fig 7). This is in line with the findings of Duncan et al. (2016). Analyzing OMI data over the period from 2005 to 2014, they conclude to a complex spatial distribution of the $NO_2$ trends characterized by a decrease in the

Seoul metropolitan area and an increase outside of the city center. The heterogeneity of changing emissions leads to a high dependence of the trend calculation across the city (change from about -30% to +10%). The radial dilution correction applied in our study, based on $NO_2$ VCD values from 2005, might therefore not be realistic enough for the complex case of Seoul. For the Beijing case, Duncan et al. (2016) also showed a reduction of the tropospheric $NO_2$ (by about -10.3% from 2005 to 2014), with a minimum in 2008 at the time of the Olympic Games.

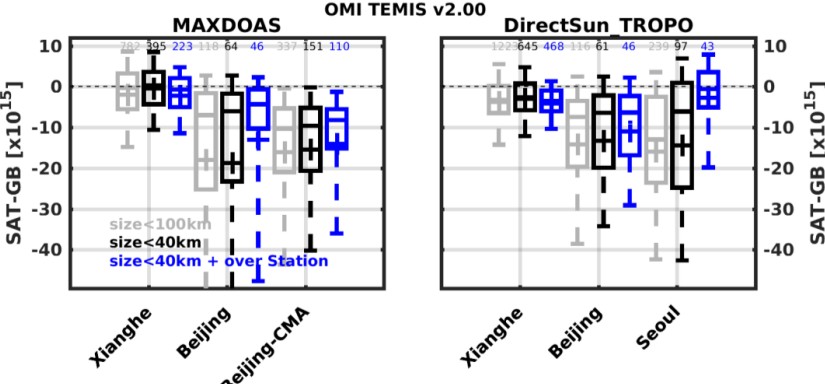

**Figure 12**: Impact of the OMI pixel size (pixels smaller than 100 km and 40 km in grey and black respectively) and with filtering on pixels only above the station (blue), on the differences deviation between satellite and ground-based data, at a few stations: Xianghe, Beijing, Beijing-CMA and Seoul. The number of comparison points is indicated on top with the corresponding colors. The box and whisker plots are defined as in Figure 10.



The peculiar behaviors of Beijing and Seoul cancel out when grouping results from all the UIPP stations. Figure 13 summarizes the change in biases for the UIPP ensemble, for the three pixels selection cases presented before. As can be seen, restricting the comparison to small pixel sizes (from 100 to 40 km) only slightly improves the median bias, but it reduces the comparison spread. When focusing on pixels in strict overpass with the stations, the bias is also reduced, but for the MAX-DOAS ensemble, not as much as when a horizontal dilution correction is applied.

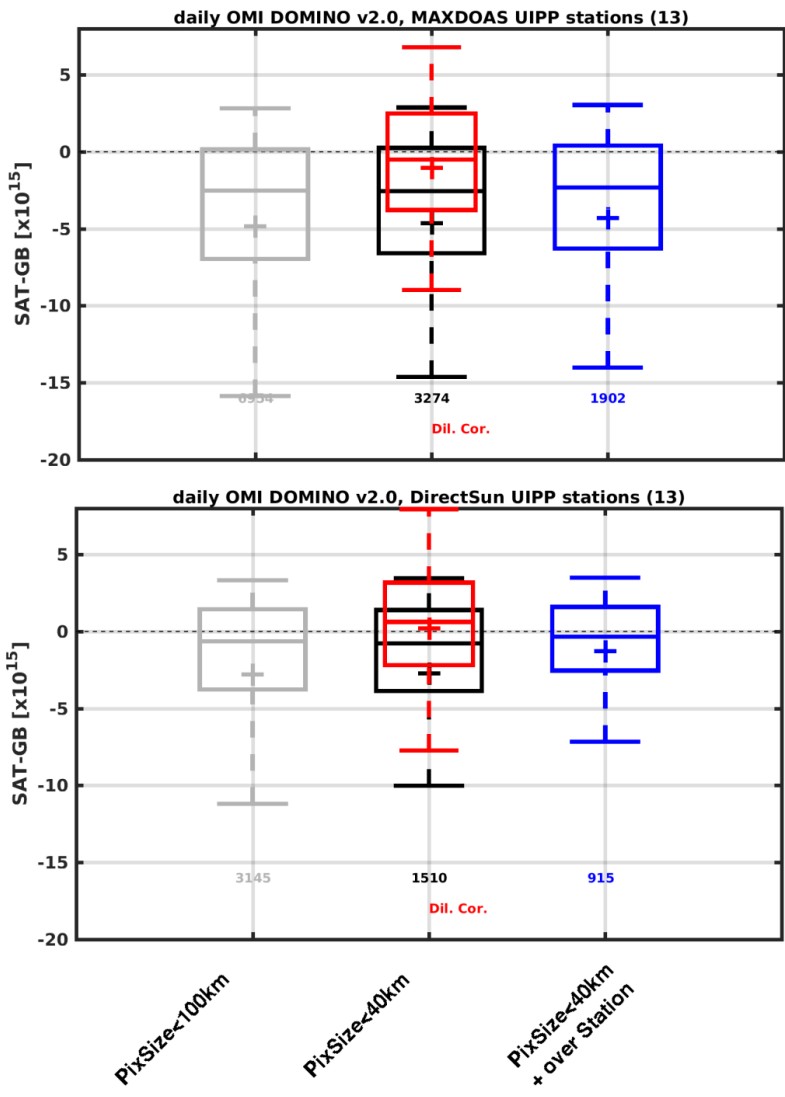



**Figure 13**: Box and whisker plot of the daily OMI DOMINO v2.0 biases for all the UIPP stations and for different possibilities of pixel size selection (pixels smaller than 100 km in grey, smaller than 4 0km in black (baseline case above) and with filtering on pixels only above the station in blue). First row: ensemble of MAX-DOAS stations, second row: ensemble of direct sun stations. The box and whisker plots are defined as in Figure 10. The number of comparison points are inserted for each cases
in the corresponding colors

## 8. Overall validation results

Figures 14 and 15 present an overview of the absolute deviations and relative differences between OMI and GOME-2A tropospheric $NO_2$ column measurements and the reference ground-based MAX-DOAS and direct-sun
measurements considered in our study. For each sensor, deviations obtained without dilution correction are presented in the upper panel (a), while biases and relative differences after application of the dilution correction are given in the lowest two panels (b and c). For panels (a) and (b), the total median instrumental errors (satellite and ground-based errors summed in quadrature) are also given as grey bars. When comparing the deviation in a) and b), the improvement by the dilution correction is clear. One can also see that results obtained using MAX-
DOAS and direct sun stations are consistent within the comparison uncertainties. Note that for a few urban sites (e.g. UHMT, Seoul), the dilution correction seems to over-correct the satellite $NO_2$ columns, especially for OMI data. This is less clear for GOME-2A, indicating that the correction approach might be slightly too aggressive for the OMI case. It can also be seen that except for a few cases, both satellite data products behave similarly at the different stations. Once corrected for the dilution effect, satellite measurements agree with ground-based data to
within 25% (black dotted lines). The blue lines represent the median bias of satellite measurements against all station data, when including the dilution correction and for ground-based VCDtropo $>2$ x$10^{15}$ molec/cm². The latter filtering is applied to remove outliers leading to unphysical mean percent values. Resulting median residual biases are -23.5% for GOME-2A and -18% for OMI. For the sake of completeness, the same analysis was also performed on QA4ECV v1.1 OMI and GOME-2A datasets, using the same selection criteria. Corresponding figures can be
found in the supplement (Fig S4 and S5). Similar results are found although the QA4ECV products tend to display slightly larger residual bias values, both for the original comparisons and after dilution correction.



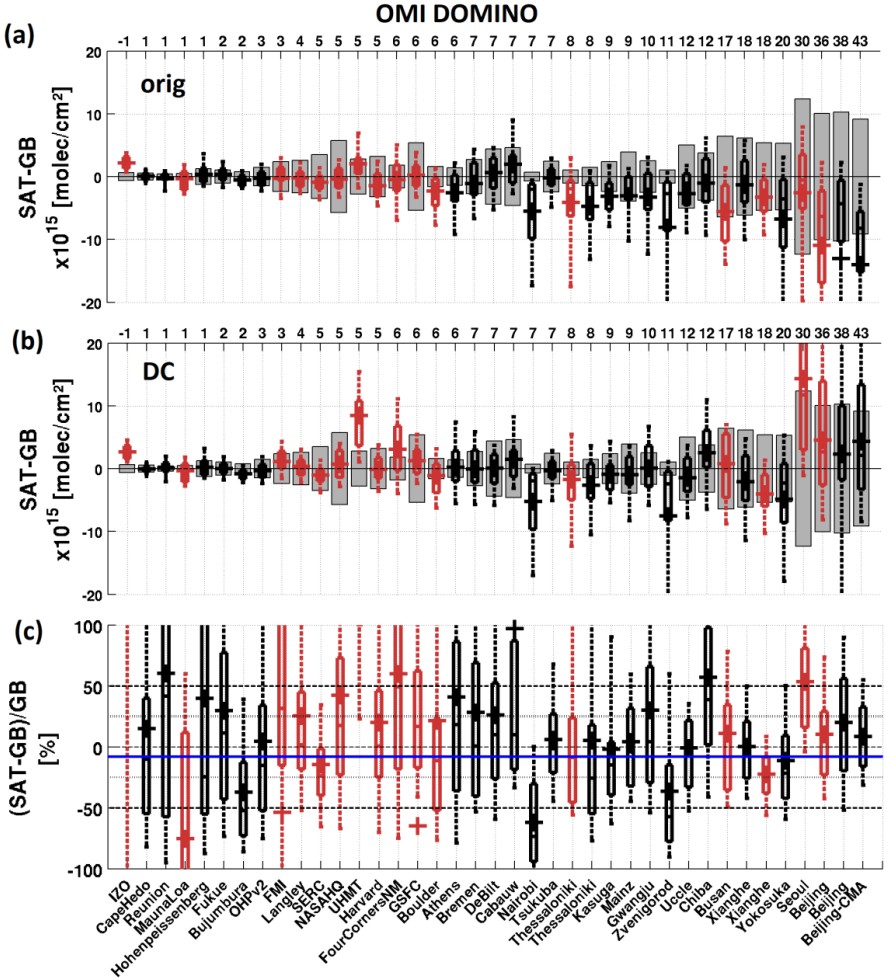

**Figure 14**: Box and whisker plot of the daily OMI TEMIS/DOMINO v2.0 biases for each station (a) for the original data, (b) and (c) when correcting for the dilution effect, in absolute and relative values. MAX-DOAS stations are presented in black, direct sun stations in dark red. The stations are ordered by increasing values of the ground-based VCDtropo, and corresponding values are given in the upper horizontal axis. The box and whisker plots are defined as in Fig 10. In panels (a) and (b), grey bars are the ± comparison error, calculated adding in quadrature the satellite and ground-based VCDtropo errors.

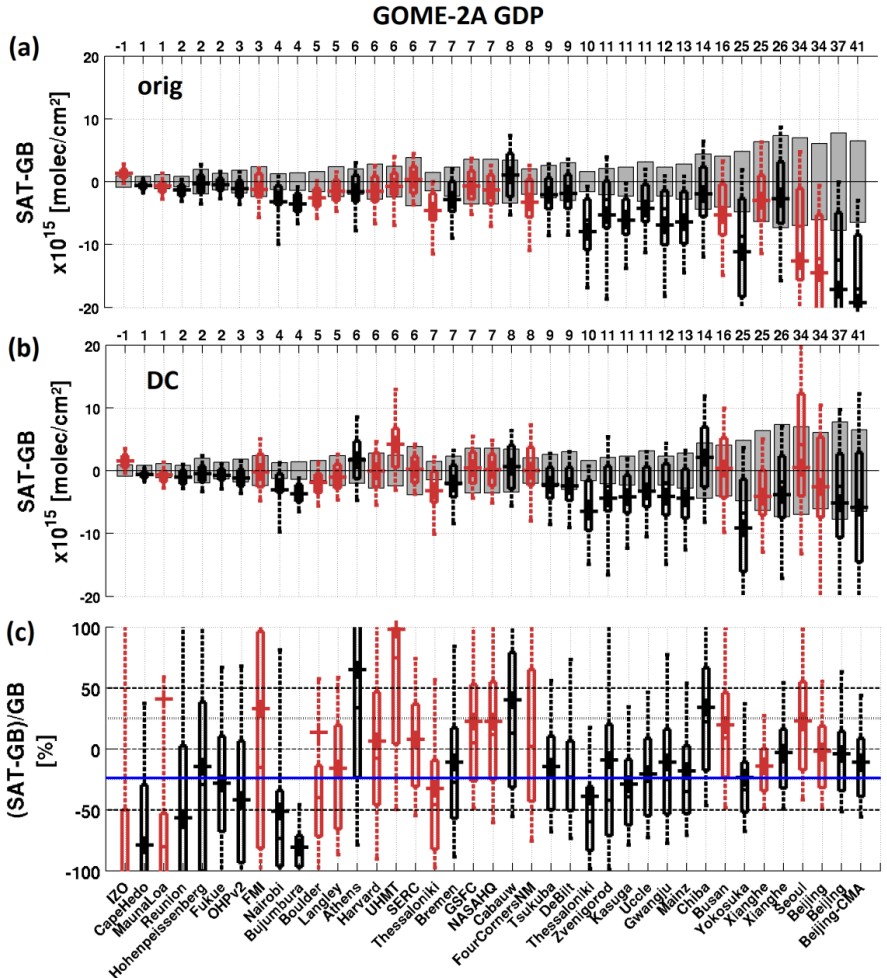

**Figure 15**: Box and whisker plot of the daily GOME-2A GDP 4.8 biases for each station (a) for the original data, (b) and (c) when correcting for the dilution effect, in absolute and relative values. MAX-DOAS stations are presented in black, direct sun stations in dark red. The stations are ordered by increasing values of the ground-based VCDtropo for the satellite overpasses coincidences, and corresponding values are given in the upper horizontal axis. The box and whisker plots are defined as in Fig. 10. In panels (a) and (b), grey bars are the ± comparison error, calculated by adding in quadrature the satellite and ground-based VCDtropo errors.

Fig 16 presents the overall GOME-2A and OMI biases for the different GDP, DOMINO and QA4ECV data products, for satellite pixels in strict coincidence with the stations. In the SAT-GB panel, grey bars present the estimated error on the median bias for each comparison case, estimated as:

$$Err = 2 * MAD / \sqrt{n} \qquad (5)$$





Where n is the number of comparisons of each case (which can be different), MAD is the median absolute deviation (see Huber (1981)), a robust indicator:

$$MAD = k * median(abs(SATi - GBi) - median(SATi - GBi)) \qquad (6)$$

k = 1.4826, for a correspondence of MAD with the 1 sigma standard deviation in case of normal distribution

without outliers. We note that the errors on the median values are significantly smaller (around $2 \times 10^{14}$ molec/cm²) than the median values themselves (a few $1 \times 10^{15}$ molec/cm²), indicating that the derived residual biases are significant.

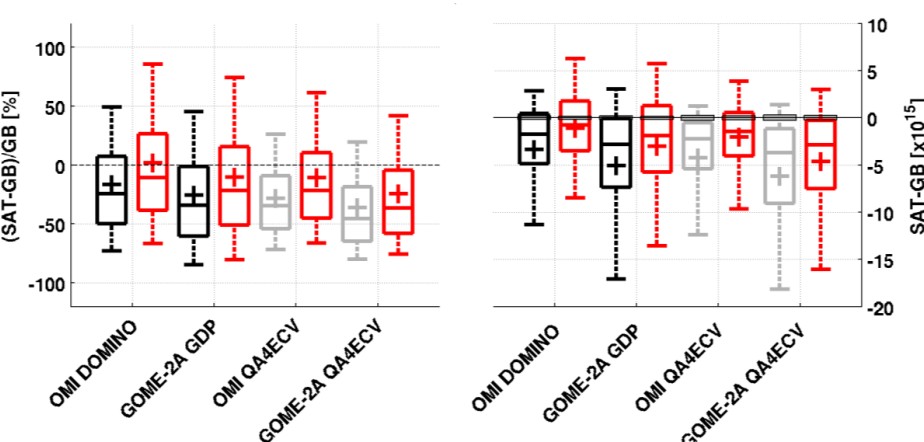

**Figure 16**: Box and whisker plot of the daily satellite biases for all stations together, in absolute and relative values. The box
and whisker plots are defined as in Fig. 10.

Table 6 summarizes the median biases for all the cases. As already stated, the dilution correction improves the validation results for both sensors, by about 10 to 13% in total over the station ensemble. The impact of considering only pixels over the stations is to reduce the bias by 2 to 6% for OMI (in comparison to the usual daily closest

pixels selection), but it has a negligible effect on GOME-2A, probably due to the large size of the GOME-2A pixels (40x80 km²). When considering the best comparison conditions including dilution correction (last column of Table 6), we come to the conclusion that satellite tropospheric $NO_2$ measurements tend to underestimate ground-based reference data by, respectively:

- 22% for GOME-2A GDP4.8,
- 36% for GOME-2A QA4ECV,
- 11% for OMI DOMINO,
- 21% for OMI QA4ECV.

It should be noted that in addition to this relative bias, the previously found positive intercepts and slopes smaller
than one (see Table 5), could point to a twofold effect, involving a multiplicative error source (e.g. the AMF) and an additive error source (e.g. the stratosphere to troposphere separation). This question should be further


investigated in future studies using more extended validation data, in particular of the stratospheric $NO_2$ column (see e.g. Compernolle et al., 2020).

**Table 6**: Daily median biases for all the stations together for the original and dilution corrected comparison for the baseline and when filtering only for pixels above the stations (in molec/cm$^2$). Values are reported after filtering out GBi values smaller than 2 x10$^{15}$ molec.cm$^3$.

|  | Original baseline | Original over stations | DC baseline | DC over stations |
|---|---|---|---|---|
| **OMI DOMINO** | -2 x10$^{15}$ [ -30 %] | -1.7 x10$^{15}$ [ -24 %] | -1.2 x10$^{15}$ [ -18 %] | -0.8 x10$^{15}$ [ -10.6 %] |
| **OMI QA4ECV** | -2.5 x10$^{15}$ [ -38 %] | -2.2x10$^{15}$ [-34.4%] | -1.8 x10$^{15}$ [-27 %] | -1.4 x10$^{15}$ [-21.5%] |
| **GOME-2A GDP** | -2.9 x10$^{15}$ [ -36 %] | -2.8 x10$^{15}$ [ -34.2 %] | -2 x10$^{15}$ [ -23.5 %] | -1.9 x10$^{15}$ [ -21.6 %] |
| **GOME-2A QA4ECV** | -3.7 x10$^{15}$ [ -48 %] | -3.7 x10$^{15}$ [-45.6%] | -2.9 x10$^{15}$ [-39 %] | -2.9 x10$^{15}$ [-36.5%] |

**9. Conclusions**

Tropospheric $NO_2$ column data from 39 ground-based remote-sensing instruments worldwide were used to validate results from GOME-2A GDP 4.8 and QA4ECV v1.1 and OMI DOMINO v2 and QA4ECV v1.1 data products. Although the ground-based retrievals are not yet fully harmonized at network level, the ground-based datasets are treated coherently for the different stations and the study illustrates the potential capacity of MAX-DOAS and direct sun network for tropospheric $NO_2$ validation. The interest of such a network resides in the large number of stations sampling different pollution levels and scenarios, corresponding to remote, sub-urban and urban conditions. Typically, sub-urban polluted stations (e.g., Xianghe) provide best conditions for the validation of satellite $NO_2$ owing to their good representativeness of the size of the OMI or GOME-2A pixel spatial extent. Validation at more remote stations can be challenging due to usually low levels of tropospheric $NO_2$, leading to difficulties in the stratosphere-to-troposphere separation step in the satellite retrieval. Other challenging cases are cities and islands surrounded by a pristine atmosphere, such as Izaña, Reunion Island, Nairobi or and Bujumbura, leading to large biases (up to ~80%) due to smearing of the local tropospheric $NO_2$ emissions content in otherwise clean surroundings.

Comparisons at urban sites or close to strong $NO_x$ sources may suffer from smoothing difference errors due to the horizontal dilution of the measured $NO_2$ field. Therefore, a quantitative correction for the dilution effect has been developed based on the spatial distribution of tropospheric $NO_2$ columns probed by OMI and averaged over one year. This dilution correction generally improves the comparison, reducing biases due to the spatial mismatch between ground-based and satellite observations. Lower biases and less scatter are obtained when considering the closest satellite pixels within a radius of 50 km from the stations. Generally OMI DOMINO v2 data agree better with ground-based data than GOME-2A GDP 4.8, especially for comparisons with MAX-DOAS data. The dilution





correction improves the station-per-station comparisons with a few exceptions, generally at remote sites with local emissions surrounded by clean areas.

Restricting the comparison to satellite pixels covering the stations further reduces the bias and spread at urban locations, and the comparison spread at sub-urban sites for OMI data. However, the largest reduction of the bias is obtained when applying the dilution correction. In terms of validation results, MAX-DOAS and direct sun measurements are found to be highly consistent, and therefore they have been used as an ensemble to assess the accuracy of GOME-2A and OMI data. Results based on this ensemble indicate that, even after correction for the horizontal dilution effect, satellite tropospheric $NO_2$ columns are systematically biased low in comparison to ground-based measurements by 22% to 36% for GOME-2A and 11% to 21% for OMI, depending on the selected satellite product. A summary of the validation results is given in Table 6.

The dilution correction developed here is parameterized according to the distance from the station and is based on one year of OMI $NO_2$ measurements (2005). This approach has several identified limitations, such as assumptions made on the radial nature of the $NO_2$ distribution around the sites and the overall applicability of the $NO_2$ field derived in 2005. Not considered here but also potentially important is the different intra-pixel dilution expected for the OMI and GOME-2A measurements. Despite its simplicity and shortcomings, our dilution correction was shown to significantly improve validation results and we anticipate that future developments will lead to further improvements. For example, possibilities exist to use estimates of the horizontal extent of MAX-DOAS measurements to improve the colocation with satellite data. MAX-DOAS instruments can also be operated in multiple azimuthal scan mode, which could be used to further refine the colocation with satellite pixels (Brinksma et al., 2008; Gratsea et al., 2016; Ortega et al., 2015; Schreier et al., 2019; Dimitropoulou et al., 2020). Finally, imaging MAX-DOAS systems such as the IMPACT instrument (Peters et al., 2019) which provides fast sampling of the full (360°) azimuthal range, may lead to significant improvements in tropospheric $NO_2$ validation close to source regions.

To further improve validation studies, information on the vertical distribution of $NO_2$ and aerosols is also needed to test the impact of a-priori assumptions in satellite data retrieval. To some extent, this can be provided by MAX-DOAS instruments making use of vertical profiling techniques for the inversion of tropospheric profiles of $NO_2$ and aerosols.

Finally, improving and further extending existing networks are essential requirements for future operational air quality satellite validation (Veihelmann et al., 2019). In this context, important steps include:

- The further development of the PGN network of Pandora instruments, to better cover source regions in all continents and in the measurement areas of all current and future satellites.
- The inclusion of MAX-DOAS instruments in the Network for the Detection of Atmospheric Composition Change (NDACC, De Mazière et al, 2018), based on ongoing efforts to harmonize retrieval methods and develop facilities for central data processing.
- The systematic adoption of harmonized uncertainty characterization and reporting, and of harmonized data reporting formats, is another crucial point for the data usage.



On this basis, it is anticipated that significant progress will be achieved in the near future towards the development of harmonized and quality-controlled global networks of UV-Vis MAX-DOAS and direct sun instruments. The development of such networks is an essential element for the validation and cross-mission consistency of the future being built atmospheric composition satellite constellation from bridging low-earth (LEO) and geostationary (GEO) orbits, in particular the ESA/EUMETSAT Copernicus Sentinel-4 (GEO) and -5 (LEO) series (planned for launch in from 2023 to 2036), the NOAA/NASA LEO Suomi-NPP/JPSS OMPS series (started in 2011, with JPSS launches planned to 2031), the CNSA LEO Geofen-5 Environment Monitoring Instrument (2018), and the geostationary missions GEMS (2020) and TEMPO (2022) developed by the US and Korea and US, respectively.

*Code/Data availability.* The datasets generated and analyzed in the present work are available from the corresponding author on request and data per station can be requested from the individual PIs.

*Author contribution.* GP and MVR planned this study. GP performed the validation, the associated investigations and wrote the manuscript. MVR and FH contributed to the scientific discussions and to the manuscript writing. NT participated to the OMI gridded maps creation. JG maintains the GOME-2 GDP station overpasses database up to date. All other co-authors provided ground-based data for the station(s) they are responsible for or support on the satellite data or the validation method. All co-authors were involved in the discussion of the results.

*Competing interests*. The authors declare that they have no conflict of interest.

*Acknowledgements*. This work has been funded by EUMETSAT through the AC SAF Continuous Development and Operations Phase (CDOP-3), and by the Belgian Federal Science Policy Office (BELSPO) via the ProDEx B-ACSAF contribution to the AC-SAF. EUMETSAT and the AC SAF are acknowledged for the production of GOME-2A GDP 4.8 data. KNMI is acknowledged for the production of OMI DOMINO v2.0 data freely available from www.temis.nl. QA4ECV data were obtained as part of the EC FP7 project Quality Assurance for Essential Climate Variables (QA4ECV, FP-SPACE-2013-1 project No 607405). The Pandora data used in this work has been obtained partly through the Pandonia Global Network (PGN) and is available publicly. Work done by HI was supported by the Environment Research and Technology Development Fund (2-1901) of the Environmental Restoration and Conservation Agency of Japan, JSPS KAKENHI (grant numbers JP19H04235 and JP17K00529), JAXA 2nd research announcement on the Earth Observations (grant number 19RT000351), and JST CREST (grant number JPMJCR15K4). Support by Coordination Funds for Promoting AeroSpace Utilization by MEXT, Japan.

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
