# Peer review of "Validation of tropospheric NO2 column measurements of GOME-2A and OMI using MAX-DOAS and direct sun network observations"

_Atmospheric Measurement Techniques, 2020_

## Referee Comment (RC1) · Anonymous Referee #1 · 11 Jun 2020

**Pinardi et al., Validation of tropospheric NO$_2$ column measurements of GOME-2A and OMI using MAX-DOAS and direct sun network observations**

**Summary**

Validation of satellite NO$_2$ measurements via aircraft or ground-station instruments is hampered by disparities in the spatial resolutions of the different measurement types. Strong localized pollution sources can be effectively sampled at a ground station but are diluted by smearing within the satellite footprint. The authors demonstrate a method to correct for this dilution, using long-term high-resolution NO$_2$ datasets to give an estimate of the spatial variability within a satellite ground pixel. Related approaches, using proxies for NO$_2$ spatial distributions have been employed in the past for limited instruments over limited regions. However, this study is valuable, because it provides a more robust test of the correction algorithm with a large number of sites and combination of ground-based MAX-DOAS and direct-sun instruments. Results show significantly improved correlations with reduced bias between satellite (OMI and GOME-2A) and the validation data. As such, it makes an important contribution and can/should be employed in future validation studies.

The paper is well-written, organized and referenced. In addition to presenting the dilution correction, it stands by itself as a good validation study. I have only a few comments and minor corrections. If addressed, I recommend acceptance in AMT.

**General comments**

(1) The dilution factor is computed in a 50 km-radius circle centered on the measurement site. This approach is convenient, but the satellite FOVs are not circular and vary widely in size. It seems that systematic errors could be large and could be mitigated using the pixel-corner coordinates in the OMI and GOME-2A level-2 files. In the conclusions, the authors acknowledge this, but more discussion should be given earlier in the text. A comparison with a dilution correction based on physical pixel dimensions for perhaps one challenging case would be useful to show whether errors in the fixed-circle assumption are significant.

(2) An error in the estimated stratospheric component of the satellite NO$_2$ is suggested as a reason for the non-zero y-intercepts in the scatter plots of Figure 9. Highly structured stratospheres – e.g. in the NASA OMI Standard Product or from assimilation (as in DOMINO) – may be locally more realistic on a given day than smoother stratospheres (e.g. from STREAM)

but may also be prone to mean systematic biases that alias some tropospheric $NO_2$ into the stratosphere and vice versa. This is discussed in the STREAM and NASA v2 algorithm papers and elsewhere. Small stratospheric errors can be amplified by the AMFs.

a) An investigation of how stratospheric aliasing may affect validation is beyond the scope of this study, as stated. But a brief comment could be included, since the OMI and GOME-2A data used here are based on assimilated stratospheres. Future examination based on STREAM would be interesting.

b) Equation (2) states that satellite-derived stratospheric $NO_2$ is subtracted from the direct-sun measurements. If the same stratosphere has been subtracted from the satellite total columns, shouldn't any stratospheric errors at least partially cancel, leading to a ~0 intercept? Are there other factors that could cause the non-zero intercepts?

c) Minor point: For the DS data, the slopes in Figure 9 show best agreement with GOME-2A for filtered, dilution-corrected data. Table 5 suggests no filtering gives better overall agreement.  Is this again an effect related to the y-intercepts?

**Specific and minor comments**

(1) Page 2, lines 19-20: "…Since the mid-1990s, $NO_2$ has been measured from space…"

(2) Page 2, line 23: "…afternoon have also been made by the OMI…"

(3) Page 4, line 26: "…from slant (SCD) to vertical (VCD) column densities."

(4) Page 4, line 37: "…2018). SCD structural uncertainties…"

(5) Page 6, line 6: "…Satellite-to-satellite comparisons…have been performed…"

(6) Page 6, line 21: "…crossing the Equator around 13:45 LT (in ascending node)."

(7) Page 6, line 29: "…the GOME-2A product…"

(8) Page 7, line 14: "…). For 18 cloud-free…"

(9) Page 8, Table 1:   The table would be easier to read if the two satellite instrument columns were better delineated (e.g. a vertical divider).  GDP4.8 and Q4ECV v1.1 should be grouped with GOME-2A and clearly marked as the first and second column headings applied to the entire table below the instrument information box. Similarly, DOMINOv2.0 and QA4ECV v1.1, grouped with OMI, should be clearly marked as the third and fourth column headings throughout.

(10) Page 14, line 33: Define DS since it is used later. "Direct-sun (DS) observations are routinely…"  Technically "direct-sun" should be hyphenated throughout, but this may be at the discretion of AMT.

(11) Page 15, line 14: "Those account for…"

(12) Page 15, line 17: "…estimated using satellite data (SAT) (alone or within assimilation…"

(13) Page 15, line 36: "…and only OMI pixels centered…"

(14) Page 16, line 4: "Ground-based (GB) MAX-DOAS date were interpolated…"

(15) Page 17, line 18: "…compared to early afternoon (13:30 hrs)…" Are LTs in this paragraph mean values for the stations (given 13:45 equator crossing)?

(16) Page 19, lines 6 and 24: "…and GOME-2A overpass…",  "…and GOME-2A overpasses…"

(17) Page 20, lines 3 and 24: "…their median difference at OMI and GOME-2A overpass are 5.7 and …",  "…than for GOME-2A…"

(18) Page 25, line 15: "…the outer extent of any 40 x 40 km$^2$ GOME-2A pixels whose centers are within the 50 km radius."

(19) Page 27, lines 3, 4: "…GOME-2A…"

(20) Page 27, line 18: "…scatter plots of GOME-2A and ground-based data…"

(21) Page 30, line 11: "…and GOME-2A GDP…"

(22) Page 31, lines 3-4: "…but should have relatively little systematic effect on regression slopes."

(23) Page 31, line 8: "…morning GOME-2A overpass…"

(24) Page 32, Figure 11: Please define seasonal colors in the caption, or preferably as a legend on the figures.

(25) Page 32, line 2: "…GOME-2A…"

(26) Page 33, line 19: "…they found a complex spatial distribution…"

(27) Page 35, line 4: "The number of comparison points for each case is shown in the corresponding color."

(28) Page 38, line 9, Figure 16 caption: Please explicitly define the colors.

---

## Referee Comment (RC2) · Anonymous Referee #2 · 1 Aug 2020

Pinardi et al present inter comparison of tropospheric $NO_2$ columns between satellite (OMI and GOME-2A) and the ground-based (direct sun and MAX-DOAS) measurements at 39 locations over a period of 2007-2018. The authors take 3 different approaches for selecting the satellite data and intercomparison: 1) all pixels within 50 km of the ground-based site; 2) only pixels smaller than 40 km and encompassing the ground-based site; 3) account for horizontal spatial heterogeneity using dilution correction derived from OMI (2005) resampled data on 0.025° x 0.025° with and without ground-based data filtering over 75th percentile. The authors presented a good literature review of the prior validation work considering spatial heterogeneity in tropospheric $NO_2$ field. They discussed in detail uncertainties in the satellite and ground-based

tropospheric $NO_2$ column retrievals. The authors concluded that satellites underestimated $NO_2$ tropospheric columns at most locations with the largest effect over the urban locations. The comparison improved if pixel size was limited and encompassed the site location. The best agreements (expressed as slopes and correlation coefficients of the linear regression analysis) were achieved from data filtering of the largest columns and applying dilution correction.

The paper is well written, addresses a very important question of satellite $NO_2$ tropospheric column quality and is within the scope of AMT.

Major comments:

I recommend the authors consider some reorganization of the paper. Based on the previous studies and the knowledge of the local sources it seems that the "base" case for the validation should be the smallest pixels encompassing the site locations and with the consideration of the measurement direction and horizontal extent within the pixels. After this comparison is done the authors can address the question of differences in pixel size and significantly reduced statistics by expending to include satellite data within 50 km of the site, demonstrating that this approach (as expected) does not improve the comparison even with the larger sample size. Then the authors can introduce the dilution correction method, which potentially increases the sample size and accounts for the heterogeneity. While this is a very promising technique especially if this can be applied to sub pixel heterogeneity, it is premature to call the dilution correction results "validation" due to correctly listed limitations. There are some filter selections and classifications that need better explanation, since a somewhat different selection criterion can potentially lead to a different conclusion.

Minor comments:

P2, l34: Pandora provides operationally only total columns of $NO_2$ and $O_3$ from the direct sun measurements

P3, l20: "direct sun measurements also match better the horizontal resolution of satellite observations". DS during the summer months or near tropics at 13:30 local time will not provide a representative horizontal resolution;

P14, l29: I would recommend: "Equipped with a 2-axis positioner, direct sun-capable DOAS instruments measure non-scattered photons. Such measurements are equally sensitive to both tropospheric and stratospheric absorptions (Figure 1b). They have a very small uncertainty in AMF, and can provide accurate total column measurements with a minimum of a-priori assumptions."

P14, l35: I would recommend: Direct sun observations are routinely available from the Pandora spectrometer instruments. A standardized Pandora network has been set-up by NASA (Herman et al., 2009, Tzortziou et al., 2014, Pandora project, https://pandora.gsfc.nasa.gov/) and expanded by ESA and LuftBlick to form the PGN (Pandonia Global Network, https://www.pandonia-globalnetwork.org/).

P15, l27: how was the cloud radiance fraction selected?

P16, l9: Do you mean to say: "On this basis, in addition to the daily comparisons at each station, corresponding monthly averages were also compared." If not, please elaborate why do you think daily data are accurate enough considering spatial and temporal variability and averaging?

P18, l4: I would recommend rephrasing: Due to different deployment strategies, the direct sun measuring instruments (especially Pandoras) were located closer...

P18, l6: I would recommend rephrasing: The MAX-DOAS ensemble of stations measured $NO_2$ total columns in the 2 to 20 x $10^{15}$ molecule/cm$^2$ range...

P18, l7-8: I am not sure how relevant this statement is to the satellite validation since accuracy of both satellite and MAX-DOAS retrievals are impacted by the clouds. A part of the observed variability in MAX-DOAS measurements is the retrieval error since most MAX-DOAS inversion algorithms assume cloud-free conditions.

P19, l7-8. Part of the bias can also be difference in $NO_2$ molecular absorption cross section temperature used in DOAS analysis. MAX-DOAS is typically analyzed using 298K while direct sun (at least Pandora data) at the profile effective temperature of 254K. Spinei et al., 2014 (https://amt.copernicus.org/articles/7/4299/2014/amt-7-4299-2014.pdf) showed that at polluted sites during hot summer months this could result in 5-10% underestimation in $NO_2$ total column derived from the direct sun data compared to the true effective temperature.

P20, l19: what definition was used for urban and suburban? Is there some specific distance and "source" size used?

P20, l19: It appears that the "goodness" of the linear correlation, as shown in Fig 5, is almost entirely driven by the highly polluted sites for GOME-2A with MAX-DOAS comparison. If for some reason Yokosuka and Beijing data were removed the conclusion about the correlation "goodness" will be very different. In my opinion, the authors did not convenience the readers that using the urban-suburban classification vs. "source strength combined with the source size" help understand actual correlation between the satellite and ground-based measurements.

P27, l22: While the slope improves, the scatter actually gets worse. Adding fit RMS might be more representative of the actual fit quality.

P28, l4-5: Pandora is a spectroscopic instrument with a 2-axis positioner, diffusers and neutral density filters to allow for a wide dynamic range measurements (direct sun, moon, and multi-axis). I would recommend changing: This is likely related to the fact that, as already mentioned, direct sun measurements (specifically Pandoras) tend to be located…Another potential reason is also higher uncertainty in determination of the "true" amount in the reference spectrum and much more "localized" measurements (e.g. at high sun)

P30, l13: Why 9th and 91th percentiles were chosen?

Fig. 11: please add the color-coding.

[Figure]

---

## Author Comment (AC1) · 5 Sep 2020

**Answer to RC1:**

Reviewer comments are given in black and author answers are in blue. Changes in the revised manuscript are marked in red.

Following a request from Referee 2 the structure of the paper has been modified. Red is used for corrections/additions, while green is used for indicating pieces of text having been moved in the paper.

**Summary**

Validation of satellite NO2 measurements via aircraft or ground-station instruments is hampered by disparities in the spatial resolutions of the different measurement types. Strong localized pollution sources can be effectively sampled at a ground station but are diluted by smearing within the satellite footprint. The authors demonstrate a method to correct for this dilution, using long-term high-resolutionNO2datasets to give an estimate of the spatial variability within a satellite ground pixel. Related approaches, using proxies for NO2 spatial distributions have been employed in the past for limited instruments over limited regions. However, this study is valuable, because it provides a more robust test of the correction algorithm with a large number of sites and combination of ground-based MAX-DOAS and direct-sun instruments. Results show significantly improved correlations with reduced bias between satellite (OMI and GOME-2A) and the validation data. As such, it makes an important contribution and can/should be employed in future validation studies. The paper is well-written, organized and referenced. In addition to presenting the dilution correction, it stands by itself as a good validation study. I have only a few comments and minor corrections. If addressed, I recommend acceptance in AMT.

**Answer:** We thank the reviewer for his useful comments and suggestions. We answer to each point below.

**General comments**

(1) The dilution factor is computed in a 50 km-radius circle centered on the measurement site. This approach is convenient, but the satellite FOVs are not circular and vary widely in size. It seems that systematic errors could be large and could be mitigated using the pixel-corner coordinates in the OMI and GOME-2A level-2 files. In the conclusions, the authors acknowledge this, but more discussion should be given earlier in the text. A comparison with a dilution correction based on physical pixel dimensions for perhaps one challenging case would be useful to show whether errors in the fixed-circle assumption are significant.

**Answer:** Another dilution correction approach has been tested, to estimate an uncertainty for our dilution method. Starting from the same high resolution OMI QA4ECV grid used for the current dilution correction, a ratio between each grid cell and the cell containing the station has been calculated. Instead of calculating a radial correction for the pixel's center distance, a correction based on the weighted value of the intersection of the high resolution grid cells ratios and the polygon formed by the satellite pixels corners is calculated, and this value is used instead of Fdil(R) in eq (4).

2 extreme cases and 2 normal cases are tested. For cases with strong over-estimation, the new method changes significantly the results, reducing the impact of the correction by about 8 to 65% (Beijing and UHMT). In the normal cases, the change is only of about 3 to 7% (Xianghe and Uccle). These changes are of about half the value of the current dilution correction for these stations. Considering the small number of extreme cases (4 over 39 cases), and the corresponding small number of comparison colocations, an uncertainty of 5% is estimated for the whole station ensemble, which is half the overall impact of the dilution (10 to 13%), as summarized in Table 6.

The outcome is discussed in Sect. 6.2 in P. 29, lines 36-39 (in addition to the existing discussion of P. 35) and in Sect. 7, P.38, lines 14-15:

*P. 29:* An alternative dilution correction approach taking into account the geographical extent of the satellite pixel and its localisation in the NO2 field has been tested. In order to estimate an uncertainty on our correction method, we applied this modified scheme to two extreme urban cases (Beijng and UHMT), and two moderate cases (Xianghe and Uccle). Differences amounting to about half the value of the current dilution correction are obtained.

P. 38: the dilution correction improves the validation results for both sensors, by about 10 to 13% in total over the stations ensemble, with an overall uncertainty due to the method, estimated to about 5%.

(2) An error in the estimated stratospheric component of the satellite NO2 is suggested as a reason for the non-zero y-intercepts in the scatter plots of Figure 9. Highly structured stratospheres –e.g. in the NASA OMI Standard Product or from assimilation (as in DOMINO)–may be locally more realistic on a given day than smoother stratospheres (e.g. from STREAM) but may also be prone to mean systematic biases that alias some tropospheric NO2 into the stratosphere and vice versa. This is discussed in the STREAM and NASA v2 algorithm papers and elsewhere. Small stratospheric errors can be amplified by the AMFs.

a) An investigation of how stratospheric aliasing may affect validation is beyond the scope of this study, as stated. But a brief comment could be included, since the OMI and GOME-2A data used here are based on assimilated stratospheres. Future examination based on STREAM would be interesting.

**Answer:** a comment on Compernolle et al. 2020 work, mentioning results with STREAM instead of a stratospheric assimilation approach for OMI, was already included in P. 31 (former page 28), lines 12-15. The following sentence is added at the end of the paragraph, according to the reviewer suggestion:

*P. 31:* Investigation of the impact of the smoother STREAM stratosphere on the tropospheric validation results is out of the scope of this study, but would be interesting as the small stratospheric errors can be amplified by the AMFs.

b) Equation (2) states that satellite-derived stratospheric  $NO_2$  is subtracted from the direct-sun measurements. If the same stratosphere has been subtracted from the satellite total columns, shouldn't any stratospheric errors at least partially cancel, leading to a ~0 intercept? Are there other factors that could cause the non-zero intercepts?

**Answer:** The satellite's tropospheric VCD retrieval is not performed from the satellite total VCD columns to get the tropospheric VCD, but from the total SCD, following equation (1). An error on VCDstrato would therefore be normalized by the satellite AMFstrato/AMFtropo ratio.

Systematic errors on the satellite slant column would lead to an additive offset in the comparisons. E.g., a wavelength calibration misfit, as shown for OMI, can have effects of NO2 slant column by  $0.85 \times 10^{15}$ molec cm-2, independent of latitude, solar zenith angle and NO2 value (van Geffen et al. 2015). This additive positive SCD offset would be present in both satellite's stratospheric SCD and in total SCD, thus cancelling itself in the satellite's tropospheric VCDtropo calculation, while it would affect the direct sun tropospheric VCD through VCDstrato substraction in equation (2).

Another factor leading to a non-zero intercept is related to the linear regression used. When the comparison points do not strictly follow a linear relationship, i.e., situations where data tend to agree well for small and intermediate values, but show large discrepancies for large values, the regression is strongly influenced by the high columns. This situation leads to a shifting of the regression line to positive y-intercepts. This is why the filtering on percentile 75 was introduced.

c) Minor point: For the DS data, the slopes in Figure 9 show best agreement with GOME-2A for filtered, dilution-corrected data. Table 5 suggests no filtering gives better overall agreement. Is this again an effect related to the y-intercepts?

**Answer:** It is unclear to us what in Table 5 suggests that "no filtering gives better overall agreement". Based on Table 5 we conclude, for the DS data, that:

- the best correlation (0.91 and 0.83), slope (1.1 for both GOME-2 and OMI) and intercept is obtained for filtered data with dilution correction (last line),

- a smaller relative bias is observed for all the data with dilution correction (and no filtering), but with large positive intercepts (between 3.2 and 3.6).

The RMS parameter has been added to table 5, following comments of referee 2. This variable also tends to be slightly larger for direct-sun dilution corrected data, confirming the over-correction. A sentence has been added:

*P.* 31, line 5: In the case of direct sun data, however, we note that the dilution correction tends to overcorrect satellite measurements (see also Fig. 9), also resulting in slightly larger RMS values for the dilution corrected cases.

**Specific and minor comments**

(1) Page 2, lines 19-20: "...Since the mid-1990s, NO2 has been measured from space..." -->done

(2) Page 2, line 23: "...afternoon have also been made by the OMI..." -->done

(3) Page 4, line 26: "...from slant (SCD) to vertical (VCD) column densities." -->done

(4) Page 4, line 37: "...2018). SCD structural uncertainties..." -->done

(5) Page 6, line 6: "...Satellite-to-satellite comparisons...have been performed..." -->done

(6) Page 6, line 21: "...crossing the Equator around 13:45 LT (in ascending node)." -->done

(7) Page 6, line 29: "...the GOME-2A product..." -->done

(8) Page 7, line 14: "...). For 18 cloud-free..." -->done

(9) Page 8, Table 1: The table would be easier to read if the two satellite instrument columns were better delineated (e.g. a vertical divider). GDP4.8 and Q4ECV v1.1 should be grouped with GOME-2A and clearly marked as the first and second column headings applied to the entire table below the instrument information box. Similarly, DOMINOv2.0 and QA4ECV v1.1, grouped with OMI, should be clearly marked as the third and fourth column headings throughout. -->done

(10) Page 14, line 33: Define DS since it is used later. "Direct-sun(DS) observations are routinely..." -->done

Technically "direct-sun" should be hyphenated throughout, but this may be at the discretion of AMT. -->done

(11) Page 15, line 14: "Those account for..." -->done

(12) Page 15, line 17: "...estimated using satellite data (SAT) (alone or within assimilation..." -->done

(13) Page 15, line 36: "...and only OMI pixels centered..." -->done

(14) Page 16, line 4: "Ground-based (GB) MAX-DOAS date were interpolated..." -->done

(15) Page 17, line 18: "...compared to early afternoon (13:30 hrs)..." Are LTs in this paragraph mean values for the stations (given 13:45 equator crossing)? --> These numbers are the solar local time at equator crossing

(16) Page19, lines 6 and 24: "...and GOME-2A overpass...", "...and GOME-2A overpasses..." -->done

(17) Page 20, lines 3 and 24: "...their median difference at OMI and GOME-2A overpass are 5.7 and ...", "...than for GOME-2A..." -->done

(18) Page 25, line 15: "...the outer extent of any 40 x 40 km2 GOME-2A pixels whose centers are within the 50 km radius." -->done

(19) Page 27, lines 3, 4: "...GOME-2A..." -->done

(20) Page 27, line 18: "...scatter plots of GOME-2A and ground-based data..." -->done

- (21) Page 30, line 11: "...and GOME-2A GDP..." -->done
- (22) Page 31, lines 3-4: "...but should have relatively little systematic effect on regression slopes."

-->done

(23) Page 31, line 8: "...morning GOME-2A overpass..." -->done

(24) Page 32, Figure 11: Please define seasonal colors in the caption, or preferably as a legend on the

figures. -->done (in the new figure 12, following the reorganization of the paper suggested by reviewer 2) (25) Page 32, line 2:"...GOME-2A..." -->done

(26) Page 33, line 19: "...they found a complex spatial distribution..." -->done

(27) Page 35, line 4: "The number of comparison points for each case is shown in the corresponding color." -->done

(28) Page 38, line 9, Figure 16 caption: Please explicitly define the colors -->an explanation of the colors has been added in the figure caption, as follows:

Red color is used for the dilution corrected data, while black is used for the previously presented products (OMI DOMINO and GOME-2A GDP) and grey is used for the QA4ECV products.

---

## Author Comment (AC2) · 5 Sep 2020

**Answer to RC2:**

Reviewer comments are given in black and author answers are in blue. Changes in the revised manuscript are marked in red.

Pinardi et al present inter comparison of tropospheric $NO_2$ columns between satellite (OMI and GOME-2A) and the ground-based (direct sun and MAX-DOAS) measurements at 39 locations over a period of 2007-2018. The authors take 3 different approaches for selecting the satellite data and intercomparison: 1) all pixels within 50 km of the ground-based site; 2) only pixels smaller than 40 km and encompassing the ground-based site; 3) account for horizontal spatial heterogeneity using dilution correction derived from OMI (2005) resampled data on 0.025_ x 0.025_ with and without ground-based data filtering over 75th percentile. The authors presented a good literature review of the prior validation work considering spatial heterogeneity in tropospheric $NO_2$ field. They discussed in detail uncertainties in the satellite and ground-based tropospheric $NO_2$ column retrievals. The authors concluded that satellites underestimated $NO_2$ tropospheric columns at most locations with the largest effect over the urban locations. The comparison improved if pixel size was limited and encompassed the site location. The best agreements (expressed as slopes and correlation coefficients of the linear regression analysis) were achieved from data filtering of the largest columns and applying dilution correction.
The paper is well written, addresses a very important question of satellite $NO_2$ tropospheric column quality and is within the scope of AMT.

Major comments:
I recommend the authors consider some reorganization of the paper. Based on the previous studies and the knowledge of the local sources it seems that the "base" case for the validation should be the smallest pixels encompassing the site locations and with the consideration of the measurement direction and horizontal extent within the pixels. After this comparison is done the authors can address the question of differences in pixel size and significantly reduced statistics by expending to include satellite data within 50 km of the site, demonstrating that this approach (as expected) does not improve the comparison even with the larger sample size. Then the authors can introduce the dilution correction method, which potentially increases the sample size and accounts for the heterogeneity. While this is a very promising technique especially if this can be applied to sub pixel heterogeneity, it is premature to call the dilution correction results "validation" due to correctly listed limitations. There are some filter selections and classifications that need better explanation, since a somewhat different selection criterion can potentially lead to a different conclusion.

*Answer: We thank the reviewer for his comments and for his suggestion to reorganize the manuscript structure. Our new baseline is now considering small pixels encompassing the site location. Since measurement pointing direction and horizontal extension are not provided for all ground-based data sets, both parameters have not been taken into account in the new baseline. The manuscript structure is reorganized as follows: Sect. 4 and 5 have been adapted for the new baseline with updates of figures 2 to 6. Figures 12 and 13 of Sect. 7.2 are shifted to a new Sect. 5.3 and become figures 7 and 8, while figures 7 to 11 become figures 9 to 12 in Sect. 6. Figures 14 to 16 do not change, and Sect. 8 becomes Sect. 7. In the revised manuscript, green is used for text moves, while red is used for other corrections/additions.*

*Regarding the measurement pointing direction and horizontal extension, a sensitivity test has been performed on Xianghe data, for which both parameters are available. Comparison with OMI has been performed by selecting all daily pixels that intersect the MAX-DOAS field of view. An average of these pixels*

*is then computed, normalized by the segment of the field-of-view crossed by each specific pixel. This comparison only shows a small increase in the number of available coincident days compared to results with only pixels over the station (from 279 to 288 coincident). The comparison results are also not significantly changed, as can be seen in Fig.1 below.*

*The following sentences were added in the paper, at P. 15, lines 30-34:*

As the pointing direction and horizontal sensitivity length are not reported for all ground-based instruments, our baseline approach is to consider only pixels encompassing the station location. However, a sensitivity test has been performed at the Xianghe station (where both parameters are provided in the data files) by selecting all pixels crossing the MAX-DOAS line of sight. Comparison results were found to be close to those from the baseline case, with only 10 additional coincident days.

[Figure]

*Figure 1. Daily and monthly comparisons at Xianghe when considering only pixels over the station (left), and pixels that crosses the MAX-DOAS field-of-view (right).*

Minor comments:
P2, l34: Pandora provides operationally only total columns of NO2 and O3 from the direct sun measurements --> *Indeed SO2 and HCHO are not "operational" products yet. We reformulated to:*
"In particular, the recently developed Pandora instrument (SciGlob, http://www.sciglob.com/) operationally provides direct sun measurements of $O_3$ and $NO_2$, and $SO_2$ and HCHO in a scientific mode.."

P3, l20: "direct sun measurements also match better the horizontal resolution of satellite observations". DS during the summer months or near tropics at 13:30 local time will not provide a representative horizontal resolution; *-->DS, by its remote sensing nature is anyway closer to the satellite measure than an in-situ measurement, but indeed, it does not "match" the satellite horizontal resolution. We reformulated to "Remote sensing measurements also match…."*

P14, l29: I would recommend: "Equipped with a 2-axis positioner, direct sun-capable DOAS instruments measure non-scattered photons. Such measurements are equally sensitive to both tropospheric and stratospheric absorptions (Figure 1b). They have a very small uncertainty in AMF, and can provide accurate total column measurements with a minimum of a-priori assumptions." *-->done*

P14, l35: I would recommend: Direct sun observations are routinely available from the Pandora spectrometer instruments. A standardized Pandora network has been set-up by NASA (Herman et al., 2009, Tzortziou et al., 2014, Pandora project, https://pandora.gsfc.nasa.gov/) and expanded by ESA and LuftBlick to form the PGN (Pandonia Global Network, https://www.pandonia-globalnetwork.org/). *-->done*

P15, l27: how was the cloud radiance fraction selected? *-->the cloud radiance fraction values are coming from the satellite retrieval, and the pixels are filtered for CRF<50%.*

P16, l9: Do you mean to say: "On this basis, in addition to the daily comparisons at each station, corresponding monthly averages were also compared." If not, please elaborate why do you think daily data are accurate enough considering spatial and temporal variability and averaging? *--> Correct, this is what it meant. The sentence has been revised as suggested.*

P18, l4: I would recommend rephrasing: Due to different deployment strategies, the direct sun measuring instruments (especially Pandoras) were located closer… *-->done*

P18, l6: I would recommend rephrasing: The MAX-DOAS ensemble of stations measured $NO_2$  tropospheric columns in the 2 to 20 x $10^{15}$ molecule/cm2 range… *-->done*

P18, l7-8: I am not sure how relevant this statement is to the satellite validation since accuracy of both satellite and MAX-DOAS retrievals are impacted by the clouds. A part of the observed variability in MAX-DOAS measurements is the retrieval error since most MAX-DOAS inversion algorithms assume cloud-free conditions. *--> MAX-DOAS retrievals assume cloud free conditions and a-posteriori cloud filtering techniques can be applied (see e.g. Gielen et al., 2014; Wang et al 2015). However, since such filtering is not applied to all stations, some datasets could still contain partially cloudy scans.*

P19, l7-8. Part of the bias can also be difference in $NO_2$ molecular absorption cross section temperature used in DOAS analysis. MAX-DOAS is typically analyzed using 298K while direct sun (at least Pandora data) at the profile effective temperature of 254K. Spinei et al., 2014 (https://amt.copernicus.org/articles/7/4299/2014/amt-7-4299-2014.pdf) showed that at polluted sites during hot summer months this could result in 5-10% underestimation in $NO_2$ total column derived from the direct sun data compared to the true effective temperature. *--> We thank the reviewer for this comment. Although it is not relevant for the discussion of Figure 4 (these three specific stations are not analyzed with cross-sections at the temperature of 254K, as done for Pandora), the following sentences have been added in the discussion of Figure 11 (former Figure 9), in P. 30:*

Potential reasons are (1) the higher uncertainty in determining the true $NO_2$ column amount in the reference spectrum and (2) the more spatially localized direct-sun measurements, especially at high sun. Moreover, the Pandora DOAS analysis is performed with $NO_2$ absorption cross section at a temperature corresponding to the effective temperature of 254K, while MAX-DOAS are typically analysed for 298K. Spinei et al. (2014) showed that at polluted sites during hot summer months this could result in 5-10% of underestimation in $NO_2$ total column derived from the direct sun data compared to the retrieval results at the true effective temperature.

P20, l19: what definition was used for urban and suburban? Is there some specific distance and "source" size used? *--> No clear definition of urban and suburban classification was found when preparing the manuscript. Thus, the classification is based on PIs knowledge of the site, communicated to the author. A clarification has been added in the text in Sect. 5.2, P. 20:*

To illustrate this point, the different stations have been qualitatively classified **by the station PIs** into urban, sub-urban and background sites (see Tables 2 and 3), based on their location with respect to known pollution sources.

P20, l19: It appears that the "goodness" of the linear correlation, as shown in Fig 5, is almost entirely driven by the highly polluted sites for GOME-2A with MAX-DOAS comparison. If for some reason Yokosuka and Beijing data were removed the conclusion about the correlation "goodness" will be very different. In my opinion, the authors did not convenience the readers that using the urban-suburban classification vs. "source strength combined with the source size" help understand actual correlation between the satellite and ground-based measurements. *--> It is indeed true that the linear regression is driven by the highly polluted sites (this is why we introduced a filtering (75th percentile) to exclude the largest columns from the linear regression, considering that extreme large values at a given site usually correspond to local events not representative of satellite observations). This is the case both for urban sites (with MAX-DOAS at Beijing and Yokosuka and direct-sun at Beijing and Seoul), and for suburban sites like Xianghe. On the other hand, our qualitative urban/suburban classification is related to the locations (and strength) of the sources from the site based on PI's knowledge, and that's why a more quantitative characterization is introduced in Sect. 6.*

P27, l22: While the slope improves, the scatter actually gets worse. Adding fit RMS might be more representative of the actual fit quality. *--> RMS values have been added for the scatter plots in tables 4 and 5.*

P28, l4-5: Pandora is a spectroscopic instrument with a 2-axis positioner, diffusers and neutral density filters to allow for a wide dynamic range measurements (direct sun, moon, and multi-axis). I would recommend changing: This is likely related to the fact that, as already mentioned, direct sun measurements (specifically Pandoras) tend to be located... *--> done*

Another potential reason is also higher uncertainty in determination of the "true" amount in the reference spectrum and much more "localized" measurements (e.g. at high sun) *--> this clarification has been added in P. 31, with also the comment on the different $NO_2$ cross-section temperature used by the Pandora and MAX-DOAS (see above).*

P30, l13: Why 9th and 91th percentiles were chosen? *--> these limits are the default values in the box-and-whisker plot routine used.*

Fig. 11: please add the color-coding. *-->done*